# Analysis of meteorological parameters triggering rainfall induced landslide: a review of 70 years in Valtellina

Andrea Abbate[1], Monica Papini[1], Laura Longoni[1]

[1] Department of Civil Engineering (DICA), Politecnico di Milano, Milano 20133, Italy

*Correspondence to*: Laura Longoni (laura.longoni@polimi.it)

**Abstract.** This paper presents an extended reanalysis of the rainfall-induced geo-hydrological events that occurred in the last 70 years in the alpine area of the Lombardy region, Italy. The work is focused on the description of the major meteorological triggering factors that have caused diffuse episodes of shallow landslide and debris flow. The aim of this reanalysis was to try to evaluate their magnitude quantitatively.

The triggering factors were studied following two approaches. The first one started from the conventional analysis of the rainfall intensity (I) and duration (D) considering local rain-gauge data and applying the I-D threshold methodology integrated with an estimation of the events' return period. We then extended this analysis and proposed a new index for the magnitude assessment (MI) based on frequency-magnitude theory. The MI index was defined considering both the return period and the spatial extension of each rainfall episode.

The second approach is based on a regional scale analysis of meteorological trigger. In particular, the strength of the extratropical cyclone structure (EC) associated with the precipitation events was assessed through the Sea Level Pressure Tendency (SLPT) meteorological index. The former has been estimated from the Norwegian Cyclone Model (NCM) theory. Both indexes have shown an agreement in ranking the event's magnitude ($R^2 = 0.88$) giving a similar interpretation of the severity that was found also in accordance with the information reported in historical databases.

This back analysis of 70 years in Valtellina identifies the MI and the SLPT as good magnitude indicators of the event, confirming that a strong cause-effect relationship exists among the EC intensity and the local rainfalls recorded on the ground. In respect to the conventional I-D threshold methodology, which is limited to a binary estimate of the likelihood of landslide occurrence, the evaluation of the MI and the SLPT indexes allow quantifying the magnitude of a rainfall episode capable to generate severe geo-hydrological hazards.

## 1 Introduction

In the context of geo-hydrological risk prevention, urban planners and infrastructure engineers still need instruments for carrying out trigger's analysis (Ozturk et al., 2015; Papini et al., 2017; Piciullo et al., 2017). This is crucial in that places

where the natural landscape has been dramatically modified by uncontrolled urbanization to avoid human injures and material damages (Albano et al., 2017b; Bronstert et al., 2018). Italy is a country historically affected by a diffuse geo-hydrological fragility (Albano et al., 2017a; Ballio et al., 2010; Caine, 1980; Gao et al., 2018; Longoni et al., 2016). Alpine and Apennines mountain slopes represent the most vulnerable places of the country where shallow landslides and debris flow can occur more frequently (Ciccarese et al., 2020; Gariano and Guzzetti, 2016; Longoni et al., 2011; Montrasio, 2000; Montrasio and Valentino, 2016; Rossi et al., 2019; Vessia et al., 2014, 2016). We can cite several examples of past events such as the case of Valtellina (Lombardy) in 1987 as well as Piedmont in 1994 and 2000 and Genova in 2011 and 2013 (Inventario Fenomeni Franosi; ISPRA, 2018). All of these catastrophic events have been caused by rather exceptional rainfall episodes that rarely occur and have particular features regards their durations and their intensities (Ceriani et al., 1994; Corominas et al., 2014; Guzzetti et al., 2007; Rappelli, 2008). Here, the scientific literature has proposed some analytical methods for relating the triggering event to the occurrence of rainfall-induced landslides.

A first methodology consists of the analysis of the rainfall return period (RP) for establishing the intensity of the meteorological trigger (Caine, 1980; Iverson, 2000). The RP has a statistical meaning and represents the average recurrence time of a rainfall episode characterized by a certain intensity ($I$) and duration ($D$), that happened at a specified location (Bovolo and Bathurst, 2012; Frattini et al., 2009; Iida, 2004). This information can potentially be linked to the recurrence of the eventually triggered geo-hydrological phenomena in case we make the hypothesis of iso-frequency with the RP of precipitation (De Michele et al., 2005; ISPRA, 2018). For a flood or a flash flood, that approximation is generally acceptable because a inundation represents the direct consequence of a heavy precipitation (Albano et al., 2017a; De Michele et al., 2005). Instead, defining a RP for a landslide is not a common practice because the failure is not a periodic event but is a sudden collapse (ISPRA, 2018). For complex and deep-seated landslides the meteorological triggering factors are also intimately bounded with the local predisposing factors, i.e. the territory morphology, geology, etc. (Ciccarese et al., 2020; Guzzetti et al., 2007; Inventario Fenomeni Franosi; ISPRA, 2018; Longoni et al., 2016; Montrasio, 2000; Ozturk et al., 2015; Papini et al., 2017). The position of the surface rupture and the seasonal groundwater circulation can have a crucial interplay role influencing the overall stability of the landslide (Longoni et al., 2014; Ronchetti et al., 2009; Xiao et al., 2020). Therefore, it is not always clear how to identify the real cause of the collapse and, the correlation with rainfall triggers is sometimes weak (Ibsen and Casagli, 2004).

A certain degree of reciprocity with precipitation triggers is maintained mainly for rainfall-induced events such as shallow landslides, soil slips, and debris flows. Therefore, a common methodology consists of the investigation of rainfall intensity-duration (I-D) curves (Ceriani et al., 1994; Ciccarese et al., 2020; Crosta and Frattini, 2001; Gao et al., 2018; Guzzetti et al., 2008; Longoni et al., 2011; Olivares et al., 2014; Peruccacci et al., 2017; Rappelli, 2008; Rosi et al., 2016; Vessia et al., 2014, 2016; Segoni et al., 2014). The rainfall thresholds are valid for a specific region where in respect to the duration and the intensity of the precipitation episode a shallow terrain movement could be triggered or not. These curves are created looking at the past events that occurred across a region, therefore, are site-specific (Ceriani et al., 1994; Guzzetti et al., 2008; Rappelli, 2008; Rossi et al., 2012). Intrinsically they include the susceptibility of the local territory to landslide failure so

their use cannot be always extended to other regions with different geological and morphological characteristics (Caine, 1980; Guzzetti et al., 2007; ISPRA, 2018; Longoni et al., 2011; Peruccacci et al., 2017; Ozturk et al., 2018). Moreover, due to their empirical nature, I-D curves are sometimes rather approximate and could detect "false alarms" or, conversely, miss some "true alarms" (Abbate et al., 2019; Guzzetti et al., 2007; Peres et al., 2018). Some studies have demonstrated their dependency on the humidity condition of superficial terrain (Jie et al., 2016; Lazzari et al., 2018). This characteristic adds further uncertainties to the reliability of I-D method. However, the I-D thresholds are widely used in the field of geo-hydrological risk prevention because permit to give a fast preliminary prediction of the occurrence of shallow soil failures in function of the local meteorological previsions (Piciullo et al., 2017).

Through the I-D threshold methodology, it is possible to distinguish critical events from non-critical ones, but no further information can be retrieved directly about their magnitude. One possible solution is to try to integrate rainfall thresholds with the probability of temporal occurrence considering again the RP of the rainfalls, under the hypothesis of iso-frequency between the triggers and the geo-hydrological effects: low magnitude events exhibit higher probabilities of occurrence while greater magnitude episodes have rare frequencies. In addition, I-D points that exhibit higher RPs are generally located at a higher distance from the I-D thresholds (Crosta and Frattini, 2001). This fact is explained by recalling the statistic of the precipitation extremes (De Michele et al., 2005) where for any fixed rainfall duration, the increase of rainfall intensity determines an increase in RP. For these reasons, that "point-threshold" distance is related to the RP and in principle could be considered for a magnitude classification of the critical event identified. Unfortunately, this assumption is generally valid only for events recorded around a very limited area where precipitation statistics are supposed to be spatial invariant.

Up to this point, we have presented the most common strategies adopted for describing the precipitation characteristics in rainfall-induced geo-hydrological events. In these methodologies only *I* and *D* parameters are investigated but are these methods enough for a complete description of the rainfall triggering factors? Is RP a good predictor of their magnitude? Can rainfall analysis be improved considering also other meteorological variables that are related to the magnitude of the trigger? In our study, we have tried to answer these questions proposing an alternative of the conventional I-D rainfall analysis be able to classify rainfall events according to the spatial extent of their impact. We propose a reanalysis of past meteorological events which provoked several landslide events. We have investigated rainfall triggers not only considering local rain gauge time-series but also including a broader description of the events looking at meteorological reanalysis maps at a regional scale. The goal was to establish a magnitude ranking among the rainfall-induced geo-hydrological events studied in order to identify the most critical ones. In this light, a 70-year reanalysis study is presented starting from a group of past rainfall episodes that happened in the alpine region of northern Lombardy, Sondrio Province, Italy (Sistema Informativo sulle Catastrofi idrogeologiche; Inventario Fenomeni Franosi; Rappelli, 2008; Tropeano, 1997). Triggering factors are interpreted following two approaches:

- In the first approach, we put the events in the context of the classical I-D approach, integrated with the estimation of RP, as mentioned earlier. We then propose an alternative for the classification of the events' magnitude through the introduction of a magnitude index (MI). The index incorporates the return period of an event with the spatial extent

of its impact in terms of landslide occurrence. The MI is defined in substitution of the classical magnitude quantification adopted for geo-hydrological events (Corominas et al., 2014; Malamud et al., 2004).

- A second approach is based on a meteorological analysis of the triggers, considering their interpretation coming from the Norwegian Cyclone Model (NCM) (Godson, 1948; Martin, 2006; Stull, 2017). Here, the trigger's magnitude is expressed through a physically-based meteorological index called the "Sea Level Pressure Tendency" (SLPT) which is a function of some atmospheric parameters evaluated at the synoptic scale and associated with the rainfall event.

To carry out our study two data sources are considered:

- Ground-based meteorological series of rainfalls (Rete Monitoraggio Idro-nivo-meteorologico; SCIA: Sistema Nazionale per l'elaborazione e diffusione di dati climatici), adopted for I-D methodology and for the RP evaluation.
- Meteorological maps provided by the National Centres of Environmental Prediction (NCEP) (Kalnay et al., 1996; Observations, Prévisions, Modèles en temps réel; National Center for Environmental Information) for the NCM intensity assessment.

The paper will be organized as follows: in section 2 is presented a brief description of the historical databases and the meteorological reanalysis maps; in section 3 are described the two methodologies behind the definition of the MI, through the extended rain-series analysis, and the SLPT, through the NCM model theory; in section 4 the outcomes from the two presented approaches are reported and the two indexes are then compared. A discussion is developed in section 5 with some comments about the obtained results, with a focus on the SLPT index performances; in the last section are reported some final remarks and conclusions of the ongoing research work.

## 2 Data and Materials

### 2.1 Historical database of geo-hydrological events and rainfall Time Series

A group of past geo-hydrological events has been considered from the alpine area of Sondrio Province, northern Lombardy, Italy, Figure 1. In our study, we have investigated historical databases to identify events that in the recent past exhibited similar cause-effect behaviour, like the 1987 event. In July 1987 this area was affected by exceptional geo-hydrological events triggered by a rather intense and prolonged rainfall episode (Rappelli, 2008; Tropeano, 1997). The effects on the territory were severe: shallow landslides, debris flows, and flash floods were recorded causing people injuries, 35 fatalities, and extended damages to infrastructure and buildings, estimated at 2 billion euros (Sistema Informativo sulle Catastrofi idrogeologiche).

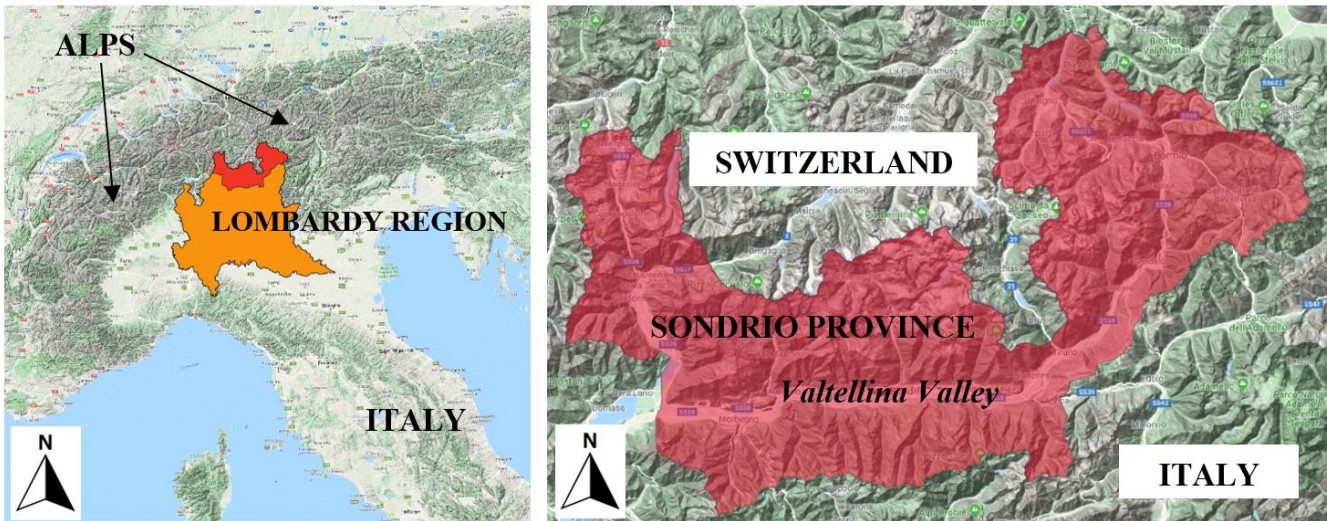

Figure 1: Case study area of Sondrio Province, northern Lombardy, base-layer from © Google Maps 2020.

Two different data sources were investigated to collect historical data: the "Aree Vulnerate Italiane" (AVI) database and the "Inventario Fenomeni Franosi Italiano" (IFFI) database (Sistema Informativo sulle Catastrofi idrogeologiche; Inventario Fenomeni Franosi). The data collect historical information from past natural disaster starting from the medieval age up to nowadays: the AVI database is directly available inside a geoportal-web site that is managed by CNR (Centro Nazionale della Ricerca) and the IFFI database, available from the national geoportal website (Sistema Informativo sulle Catastrofi

idrogeologiche; Inventario Fenomeni Franosi). Available events time-series were not homogeneous so that the consistency of the database was evaluated, redundant records have been dropped and a final integration between the AVI and the IFFI database information was carried out.

The period chosen for the reanalysis is comprised between 1951 and 2019. Systematic monitoring of the precipitation and temperature started in Italy in 1951 by SIMN (Servizio Idrografico e Mareografico Nazionale) and looking at the antecedent

periods these data were missed or characterized by several uncertainties or errors (SCIA: Sistema Nazionale per l'elaborazione e diffusione di dati climatici). The available rain gauge data series were gathered from local archives of SIMN (SCIA: Sistema Nazionale per l'elaborazione e diffusione di dati climatici) and ARPA Lombardia (Agenzia Regionale per la Protezione dell'Ambiente) (Rete Monitoraggio Idro-nivo-meteorologico). These series were conventionally recorded on daily basis until the 2000s years so "a daily rain" represents the maximum resolution of our dataset before that period.

Starting from 2001, the available temporal resolution has moved to a sub-hourly time-step increasing the accuracy of the rainfall analysis.

In AVI and IFFI database, the precise location of geo-hydrological episodes was not available even for the most recent events that happened after the 2000s. Therefore, some indications about locations were retrieved from the AVI database considering the municipalities affected by disasters. The spatial extension of affected areas (AA) describes those locations

that have experimented with some damages due to geo-hydrological events that occurred. This information is indicative of

the area where the rainfall event has been supposed to be more intense. In fact, AA was then compared with ground-based rain gauge series from the entire Sondrio Province with the aim to reconstruct for each rainfall event its spatial distribution. Selected events have been classified in function of AA parameter: extremely localized events (EXL), with an influence area lower than 1000 $km^2$, or diffuse events (DIF), with significant territorial diffusion greater than 1000 $km^2$. This threshold has been motivated referring to the nature of the meteorological triggers: EXL were generally associated with convective rainfall phenomena which extension has an order of 10 x 10 $km^2$ and DIF were characterized by diffuse and uniform rainfall with an extension around 100 x 100 $km^2$ (Martin, 2006; Rotunno and Houze, 2007). In Table 1 is reported the list of geo-hydrological events analysed in our study.

Table 1: Geo-hydrological events recorded from 1951 up to 2019 considered for the back-analysis study. In the table are also reported the event classification considering their spatial extension; the extreme localized events (EXL) and the more diffuse ones (DIF). (*) uncertainty data.

| YEAR | START | END | METEO TYPE | EFFECTS | MUNICIPALITY AFFECTED | AFFECTED AREA km² | EXTENSION TYPE | CUMULATED RAIN mm | EVENT DURATION hours |
|---|---|---|---|---|---|---|---|---|---|
| 1951 | 7 august | 8 august | Heavy rainfall | Flash Floods | 4 | 800 | EXL | 218.0 | 48 |
| 1953 | 17 july | 18 july | Heavy rainfall | Flash Floods | 3 | 250 | EXL | 83.8 | 24 |
| 1960 | 15 september | 17 september | Heavy rainfalls | Landslide and Floods | 17 | 1500 | DIF | 115.6 | 48 |
| 1966 | 3 november | 5 november | Prolonged rainfalls | Landslides and Floods | 3* | 1000 | DIF | 128.6 | 72 |
| 1983 | 21 may | 23 may | Heavy rainfalls | Landslides | 12 | 500 | EXL | 208.6 | 72 |
| 1987 | 16 july | 19 july | Prolonged rainfalls | Landslides and Floods | 77 | 3000 | DIF | 254.8 | 96 |
| 1997 | 26 june | 29 june | Prolonged rainfalls | Landslides and Floods | 6 | 500 | EXL | 275.0 | 96 |
| 2000 | 13 november | 17 november | Prolonged rainfalls | Landslides | 60 | 2000 | DIF | 218.7 | 96 |
| 2002 | 13 november | 18 november | Prolonged rainfalls | Landslides | 60 | 2000 | DIF | 308.8 | 120 |
| 2008 | 12 july | 13 july | Heavy rainfalls | Landslides | 6 | 300 | EXL | 60.0 | 12 |
| 2018 | 27 october | 30 october | Prolonged rainfalls | Landslides | 20 | 1500 | DIF | 242.4 | 96 |
| 2019 | 11 june | 12 june | Heavy rainfall | Landslides and Floods | 9 | 700 | EXL | 110.0 | 13 |

## 2.2 NCEP reanalysis maps

To improve the description of rainfall triggering factors, the meteorological reanalysis maps were examined considering the National Centre for Environmental Prediction (NCEP) data (Kalnay et al., 1996; Observations, Prévisions, Modèles en temps réel). The former has a spatial resolution of 2.5° x 2.5° degrees of latitude and longitude, covering the whole planet with a

temporal frequency of 12 h. All the data stored in NCEP maps are useful for the interpretation of air masses dynamic in the middle latitudes such as the Extratropical Cyclones (ECs) that are responsible for the spatial and temporal evolution of intense precipitation phenomena. For the European region, ECs develops in the Atlantic Ocean near the British Islands. ECs

are deputed for the large part of precipitation recorded over the Alps mountain range (Rotunno and Houze, 2007) because they are generally advected eastward through the Mediterranean area by Rossby waves (RW) (Grazzini and Vitart, 2015; Martin, 2006). At the boundary of the polar vortex, RW can generate strong jet streams that can move air masses in the direction of the southern Alps, enhancing vertical air motions. Across the southern flank of the Alps, this mechanism may lead to trigger persistent and heavy precipitation (Rotunno and Houze, 2007) that can intensify if an orographic uplift of the

incoming southerly flow is also triggered (Abbate et al., 2021; Grazzini, 2007). Rainfall can reach remarkably high amounts if these conditions are prolonged for several days, leading up to 400 mm in 2/3 days (Grazzini, 2007; Rotunno and Houze, 2007). For each event listed in Table 1, we have examined correspondent NCEP maps to investigate the mechanism responsible for generating those intense precipitations over the target area.

## 3. Model and Methods

The triggers analysis is here presented considering the I-D thresholds approach, its extension through the MI index definition and the NCM model with SLPT index evaluation.

### 3.1 Rainfall I-D thresholds and Return Period analysis

The daily rainfall rate has been determined from the total amounts and the duration listed in Table 1. Rainfall amounts (RA) were estimated keeping the distinction between EXL and DIF events, Figure 2.A. For EXLs the nearest rain gauge or at least

the 2 nearest rain gauges were chosen as reference. For DIFs, all the available daily rain data $RA_i$ in the territory have been summed and averaged considering the number of rain gauges stations "$n$" to obtain a representative value for $RA_{avg}$, Eq. (1.a). We have assumed the hypothesis of a uniform spatial distribution of the rain gauge stations, Figure 2.B, neglecting any influence of elevation on rainfall data (Abbate et al., 2021). Then, RR was computed as the ratio of the cumulative rainfall $RA_{avg}$ on the duration $D$, Eq. (1.b).

$$RA_{avg} = \frac{\sum_{i=1}^{n} RA_i}{n} \quad in\ mm \tag{1.a}$$

$$RR = \frac{RA_{avg}}{D} \quad in\ mm\ h^{-1} \tag{1.b}$$


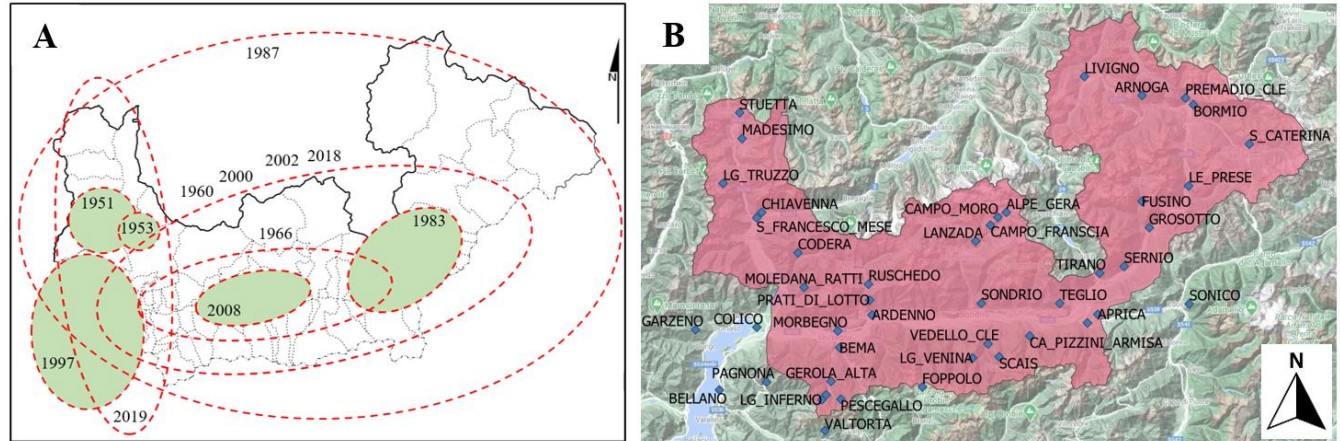

Figure 2: A) distribution of critical events across Sondrio province and B) the local rain gauge station network considered in the study, base-layer from © Google Maps 2020.

For the studied area, a set of thresholds proposed in the literature was considered, reported in Table 2. All the rainfall
thresholds have a monomial expression where $D$ is the duration of the rainfall (hours), and $I$ is the average rainfall intensity
(mm h$^{-1}$). The "Caine" curve (2.a) (Caine, 1980) is the most general one, valid worldwide for shallow landslides and debris
flow phenomena. At a regional scale, a more recent study conducted by (Guzzetti et al., 2007) proposed a new set of curves
valid for central and southern Europe, considering a distinction among different climate types. In our study, three of them
were selected: the general one (2.b), the curve valid for mid-climate (2.c), and the one suitable for highlands and mountain
environments (2.d). Another study from (Peruccacci et al., 2017) further extended the previous study by Guzzetti addressing
a new I-D threshold valid for the Italian country. At the local scale, the "Cancelli Nova" (2.e) (Rappelli, 2008), the
"Ceriani" (2.f) (Ceriani et al., 1994), and the "Crosta Frattini" curves (2.h) (Crosta and Frattini, 2001) were proposed
respectively in 1985, 1994 and 1998. All of them were calibrated directly on the recorded data available in Sondrio Province.

Table 2: I-D threshold curve set considered in this study.

| I-D THRESHOLD CURVES | AUTHORS | VALIDITY | EQUATION |
|---|---|---|---|
| $I = 14.84\,D^{-0.39}$ | Caine | World | (2.a) |
| $I = 8.67\,D^{-0.61}$ | Guzzetti | Regional (Italy) | (2.b) |
| $I = 18.6\,D^{-0.81}$ | Guzzetti | Regional (Italy) | (2.c) |
| $I = 8.53\,D^{-0.64}$ | Guzzetti | Regional (Italy) | (2.d) |
| $I = 7.70\,D^{-0.39}$ | Peruccacci | Regional (Italy) | (2.e) |
| $I = 44.67\,D^{-0.78}$ | Cancelli Nova | Local (Lombardy region) | (2.f) |
| $I = 20.01\,D^{-0.55}$ | Ceriani | Local (Lombardy region) | (2.g) |
| $I = 12\,(D^{-1} + 0.07)$ | Crosta Frattini | Local (Lombardy region) | (2.h) |

For each event, the couple points *RR-D* were plotted against the I-D threshold curves, and their return period RP was evaluated. The former was determined following the methodology based on the IDF curves (Intensity Duration Frequency) (De Michele et al., 2005) available for the Lombardy region and provided by (Rete Monitoraggio Idro-nivo-meteorologico).

The coefficients of IDF curves are estimated through the analysis of rainfall extremes addressing the GEV (Generalized Extreme Value) distribution. The dataset considered for the GEV was the SIMN timeseries (SCIA: Sistema Nazionale per l'elaborazione e diffusione di dati climatici) gathered from 1960 up to 1990 across the whole territory of the region. Bearing in mind that our localized events EXL has been distinguished separately in respect to the diffuse DIF, also for the RP calculation, we have considered the same assumptions as for RR evaluation. For the localized events, the on-site coefficient

of IDFs has been taken, while for the diffusive ones, a spatially averaged value has been computed.

## 3.2 Trigger's hazard estimation and the MI magnitude index

A further step in the precipitation analysis consists of the hazard and magnitude assessment for each event. According to (Guzzetti et al., 2005) the general landslide hazard could be defined as a probabilistic function of three terms Eq. (2.a): the size $A_l$, the temporal occurrence $T_1$ and the spatial susceptibility S. The "size" term has stored the information about the

volume, the area or the density of landslides occurred over a particular area. The temporal occurrence considers the periodical reactivation of a single landslide (deep-seated) or the recurrence of shallow landslides inside a catchment. The spatial susceptibility represents the quantification of the territory predisposition to a landslide phenomenon.

$$H_{landslide} = P(A_l \geq a_l) \cdot P(T_l \geq t_l) \cdot S \qquad \text{2.a}$$

$$S = 1 \qquad \text{2.b}$$

$$log(P(A_l \geq a_l)) = a - b\,AA \qquad where\ a, b\ are\ the\ coefficients \qquad \text{2.c}$$

$$log(P(T_l \geq t_l)) = c - d\,RP \qquad where\ c, d\ are\ the\ coefficients \qquad \text{2.d}$$

Starting from the definition of Eq.2.a. we have extended this concept and adapted it to interpret the events in our reanalysis

study. The aim was to define a proper hazard and then a magnitude indicator for geo-hydrological events considering the temporal and spatial probability of occurrence of the triggering rainfalls. According to (Malamud et al., 2004) a scale for the magnitude is necessary to interpret quantitatively the episodes and to highlight the most severe ones. For landslides and rainfall-induced geo-hydrological events, a unique method that describe the "energy" does not exist because several variables may play an important role in its definition (Bovolo and Bathurst, 2011; Frattini et al., 2009; Gao et al., 2018; Iida,

2004; Reid and Page, 2003). Therefore, under some hypothesis, we have proposed a new magnitude index (MI) as a quantitative parameter for assessing a proper magnitude ranking. Firstly, we have assumed that the investigated area had a homogeneous susceptibility $S = 1$ to shallow landslide and debris flow triggering. This choice was motivated by geological

and morphological features, also looking at recent susceptibility maps proposed by (ISPRA, 2018). Then we moved on other terms trying to determine the spatial and temporal probability of exceedance from AA and RP parameters, recalling the theory of frequency-magnitude relationship.

The frequency-magnitude curve (FMC) was proposed by (Gutenberg and Richter, 1944) for earthquake studies and then was also extended for interpreting different types of natural phenomena (Gao et al., 2019). The MCF curve is obtained by plotting incremental frequency $F_i$ against the magnitude $M_i$ on a logarithmic scale. $F_i$ represents the frequency of the event that has a magnitude $\geq$ of a certain value $M_i$. In our study, the MFCs were considered to evaluate the probability of occurrence of a certain event in time and space and then combined to determine its hazard as described in Eq. (2.a). The temporal occurrence term requires the estimation $P(T_l \geq t_l)$ from RP's frequency-magnitude relationship. This represents the probability of occurrence of an event $T_l$ with a RP $\geq t_l$. According to (Guzzetti et al., 2005), the other hazard component is addressed by the landslide size, Eq. (2.d). In this regard, in our database was not possible to retrieve enough information about event features, such as the volumes and areas involved or the numbers of landslide failures. Therefore, the AA parameter was used as a proxy of the "trigger's size" and was treated similarly to the RP term. The probability of spatial occurrence $P(A_l \geq a_l)$ of an event $A_l$ with a AA $\geq a_l$, was retrieved from FMC, Eq. (2.c). Then, the hazard has been estimated using the Eq. (3.a). Due to the modification of the first term $P(A_l \geq a_l)$ it not properly represents the landslide hazard, but $H_{trigger}$ is an indicator of the hazard as a function the trigger's temporal frequency and spatial extension.

In most natural cases, the frequency of low magnitude geo-hydrological events is rather high and vice-versa. Therefore, we tried to estimate the trigger magnitude as an inverse function of the hazard. The former is a combination of two probabilities of occurrence Eq. (3.b), therefore it can be transformed into a magnitude recalling again the FMC in Eq. (3.c). Working out some algebra with Eq. (3.a, 3.b and 3.c) we have obtained a representation of the magnitude expressed by the index MI, Eq. (3.d). The MI is a sum of two contributes: the first describes its spatial extension through the parameter AA and the second its temporal occurrence through the RP. In this light, the MI was intended to be more complete rather than the single RP because through AA term it is possible to consider the "integral effects" related to the trigger's extension. The MI was taken as a reference for testing the SLPT index presented in the next section.

$$H_{trigger} = P(A_l \geq a_l) \cdot P(T_l \geq t_l) \qquad\qquad 3.a$$

$$H_{trigger} = exp(a - b\,AA) \cdot exp(c - d\,RP) \qquad\qquad 3.b$$

$$M_{trigger} = -log(H_{trigger}) = -log(\,exp(a - b\,AA) \cdot exp(c - d\,RP)) \qquad\qquad 3.c$$

$$MI = M_{trigger} = -((a - b\,AA) + (c - d\,RP)) \qquad\qquad 3.d$$

## 3.2 NCM model and SLPT index

The extratropical cyclone dynamic influences the rainfall intensities: if the EC is stronger, more precipitation is expected over an area but, depending on EC spatial and temporal evolution, rainfalls could exhibit different total amounts and duration. Therefore, using the NCEP maps, the Norwegian Cyclone Model (NCM) (Godson, 1948; Martin, 2006; Stull, 2017) was chosen for estimating a strength index of ECs. NCM was formulated in the early 20th century. It describes an extratropical cyclone that develops as a disturbance along the boundary (front) between the polar and mid-latitude air masses. The model calculates indirectly the Sea-Level Pressure Tendency (SLPT), the time-variation ratio of sea-level atmospheric pressure $\Delta p_{sl}/\Delta t$ (hPa h$^{-1}$) that represents an indicator of the strength of a cyclone structure (Andrews, 2010; Godson, 1948; Martin, 2006; Stull, 2017; Wallace and Hobbs, 2006). When the EC is more intense, the absolute value of the SLPT ratio is higher and, consequently, the EC can cause more rainfalls. According to (Stull, 2017), this index is obtained as a sum of four different influencing factors that correspond to the processes implicated in the dynamic evolution of extratropical cyclone:

$$\frac{\Delta p_{sl}}{\Delta t} = T_1 + T_2 + T_3 + T_4 = SLPT \tag{4}$$

- $T_1$ expresses the "upper layer divergence mechanism" due to jet streams, which removes air mass from the air column. In the Eq (4.a2), $\rho_{MID} = 0.5 \text{ kg m}^{-3}$ is the average density of air column and $g = 9.8 \, m \, s^{-2}$. $W_{MID} \, (m \, s^{-1})$ is the mean air column vertical velocity that is evaluated considering the Eq. (4.a1) in the proximity of the local change of jet stream velocity gradient $\Delta W_{js}(m \, s^{-1})$, where $\Delta z \cong 5000$ m and $\Delta s_1$ (m) jet streak elongation. According to (Stull, 2017), Eq. (4.a1) is a strong approximation because supposes air density constant over air column, so that we have considered a revised version (Stull, 2017) that expresses the $W_{MID}$ in function of other parameters such as the geostrophic wind velocity $G(m \, s^{-1})$, the curvature radius $R \, (km)$ of Rossby waves and Coriolis parameter $f_c(s^{-1})$;

$$W_{MID} = \frac{\Delta W_{js}}{\Delta z/\Delta s_1} \tag{4.a1}$$

$$T_1 = - \, g \, \rho_{MID} \, W_{MID} \tag{4.a2}$$

- $T_2$ is the "atmosphere boundary layer pumping", which causes the horizontal wind to spiral inward toward a low-pressure centre. In Eq. (4.b2), the air density of the boundary layer is $\rho_{BL} = 1.112 \, kg \, m^{-3}$. $W_{BL}(m \, s^{-1})$ is the vertical velocities at boundary-layer calculated through Eq. (4.b1) following the approach proposed by (Stull, 2017) for cyclone structures: the $b_{BL}$ factor is a function of boundary layer thickness, that can be assumed equal to 1000 m on average, and drag coefficient $C_d \approx 0.005$ is defined for flow over land;

$$W_{BL} = \frac{2\, b_{BL}\, C_d}{f_c} \frac{G^2}{R} \qquad (4.b1)$$

$$T_2 = g\, \rho_{BL}\, W_{BL} \qquad (4.b2)$$

285

- $T_3$ expresses the horizontal air mass advection that moves a low-pressure centre in the direction of the target region, Eq. (4.c2). The advection velocity $M_c$ $(m\ s^{-1})$ is a function of the celerity of Rossby waves $c_{RW}$ $(m\ s^{-1})$ and the geostrophic wind $G$, Eq. (4.c1). The spatial pressure gradient at sea level is evaluated considering the distance $\Delta s_2 (m)$ between the low-pressure centre and the target region;

$$M_c = c_{RW} - G \qquad (4.c1)$$

$$T_3 = -\, Mc\, \frac{\Delta p_{GW}}{\Delta s_2} \qquad (4.c2)$$

290

- $T_4$ is the "latent heating" due to water vapor condensation in rainfall. It comes from the theory of thermodynamic transformations of water vapour in the atmosphere where all the parameters for rain condensation processes are stored in the term $b_{AD}$ . The precipitation that does reach the ground is related to the net amount of condensational heating during time interval $\Delta t$ by Eq. (4.d1) where $T_v$ $(K)$ is average air-column virtual temperature, $a =$ 295    $10^{-6}\ km\ mm^{-1}$, $\Delta z$ $(km)$ is the depth of the air column, the ratio of latent heat of vaporization to the specific heat of air is $Lv/Cp = 2500\ K\ kg_{air}\ kg_{liq}^{-1}$, and where $\rho_{air}$ and $\rho_{liq}$ are air and liquid-water densities, respectively, with $\rho_{liq} = 1000\ kg\ m^{-3}$. The hypsometric equation relates to pressure-temperature changes as reported in Eq (4.d2). For an air column with an average virtual temperature of $Tv \approx 300\ K$, we obtain $b_{AD} = 0.082\ kPa\ mm^{-1}$ in Eq (4.d3) that is considered for the description of the net column-average effect.

$$\frac{\Delta T_v}{\Delta t} = \frac{a}{\Delta z} \frac{L_v}{C_p} \frac{\rho_{liq}}{\rho_{air}} RR \qquad (4.d1)$$

$$\frac{\Delta P_s}{\Delta t} = -\frac{g}{T_v} \frac{L_v}{C_p} \rho_{liq} RR = -b_{AD}\ RR \qquad (4.d2)$$

$$T_4 = -b_{AD}\ RR \qquad (4.d3)$$

300

When the balance in Eq. (4) is negative the cyclogenesis occurs. $T_1$, $T_3$, and $T_4$ bring a negative contribution to strengthening the EC cyclogenesis and lowering the SLPT index. Instead, $T_2$ has a positive contribution and tends to weaken the ECs structure increasing SLPT value. In Figure 3.A and Figure 3.B have depicted the mechanisms described by four terms $T_i$. Figure 3.C reports how the model works considering the contribution of each four components across the timeline (A to G) 305    that represents the sages of EC: EC's formation phase (i.e. cyclogenesis) is from A to D stages and EC's dissipation phase (i.e. cyclolysis) is from D to G. The critical phase of the EC is in the proximity of point D where negative terms overcome the positive one. The SLPT index has been evaluated in correspondence with C / D stages.

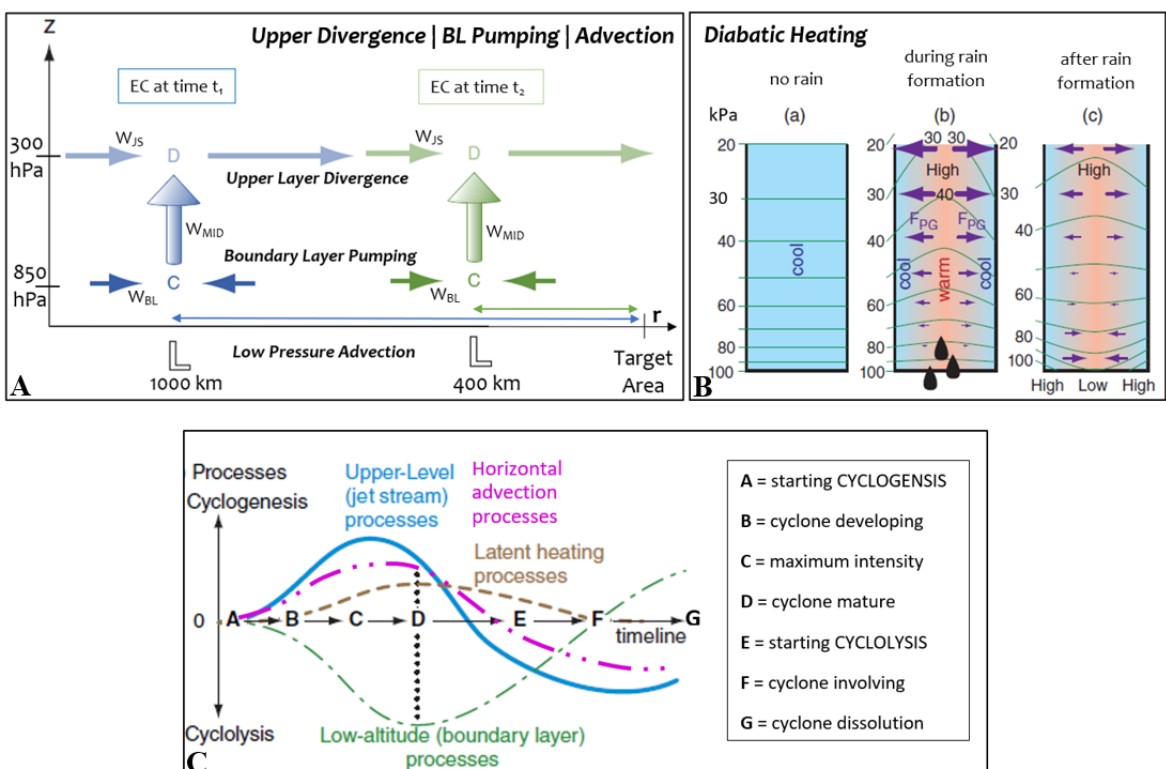

Figure 3: A) scheme of the mechanism represented by $T_1$, $T_2$, and $T_3$ terms, B) scheme of the mechanism represented by $T_4$ term, and C) qualitative temporal evolution of each of four terms $T_1$, $T_2$, $T_3$, and $T_4$ during cyclone phases (A to G) and their contribution to cyclone formation (cyclogenesis) and cyclone dissolution (cyclolysis), modified after (Stull, 2017),.

## 4 Results

In this section, the results are presented in four steps. Firstly, the qualitative analysis coming from the direct interpretation of database and NCEP maps is reported. Secondly, the I-D rainfall analysis is carried out and the MI index evaluation is described. Thirdly for each considered event, the SLPT is estimated and then compared with the MI index.

### 4.1 Database interpretation and NCEP maps

The dataset of Table 1 shows a clear seasonal distribution of the events mainly concentrated during the summer and autumn seasons. July and November are the months more prone to geo-hydrological events and this strong seasonality highlights that the triggers phenomena involved may have different origins (Martin, 2006; Rotunno and Houze, 2007). In July, meteorological events are characterized mainly by high intensity and short duration with a typical convective behaviour of precipitation (thunderstorms), and their average duration is generally around 1 or 2 days. In particular, 1951, 1953, 1987, 1997, 2008 and 2019 events happened during the summer season and rainfall cumulated were comprised between 100-200 mm, apart from 1987 and 1997 that were rather exceptional (254 mm and 275 mm in three days ). During October and

November, rainfall events are characterized by higher persistency (4-5 days) and rainfall cumulated can easily reach amounts around 250-350 mm, such as for the events that happened in 2000, 2002, and 2018.

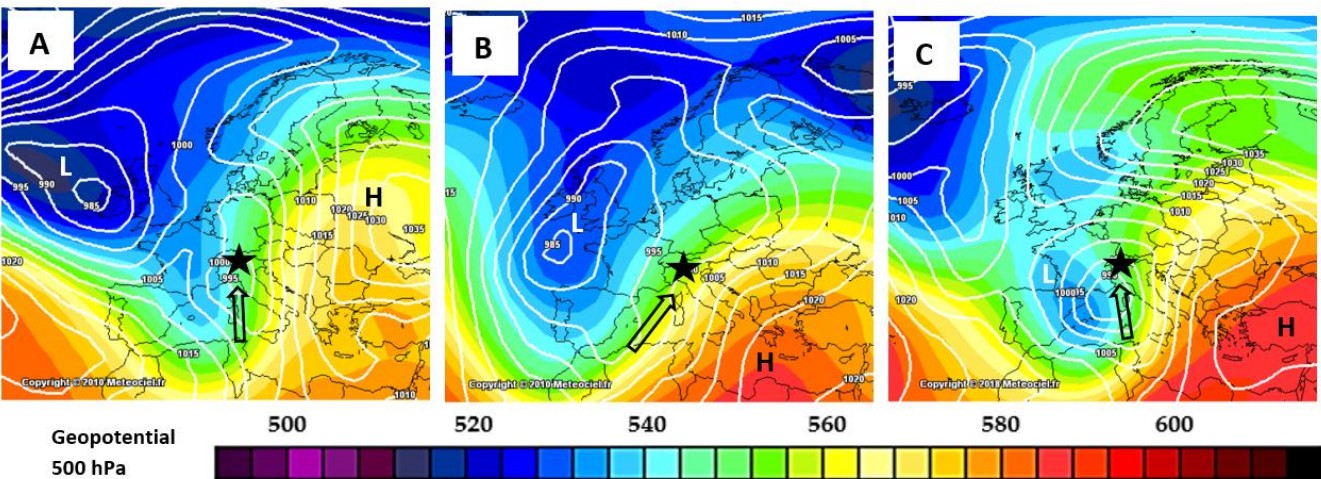

Figure 4: Reanalysis maps from NCEP reporting the Sea-Level Pressure and 500 hPa Pressure (colours) for 1966 A), 2002 B) and 2018 C) events where the black star is the Sondrio Province position, and the black arrow indicated the incoming southerly flow responsible of huge precipitation enhancing, adapted from (Meteo Ciel, 2020), modified after

Through the analysis of NCEP maps, we have observed that all the events reported in Table 1 have been triggered in correspondence of EC structures that moved eastward from the Atlantic Ocean in the direction of the Alpine mountain range. In Figure 4 are reported three examples of reanalysis maps that show the pressure distribution at 500 hPa reference height across Europe during the 1966, 2002 and 2018 events. A qualitative comparison among the three maps highlights that three events have been characterized by the evolution of a rather intense EC that is recognizable from the deep low pressure (L) located near the British Islands. This recurrent configuration has been responsible for the torrential rainfall recorded in the Southern Alps across the Sondrio Province. Consequently, the geo-hydrological effects could be directly imputed to the intensification of these EC structures. Starting from this qualitative evidence we have moved to a quantitative analysis following the two approaches proposed.

### 4.2 Approach 1: I-D threshold rainfall analysis and MI index extension

The average daily rain rate $I$ and the duration $D$ of the rainfall episodes in Table 1 were plotted against the rainfall threshold curves listed from Eq. (2.a) to Eq. (2.f) inside Figure 5. Most events can be clustered in the right-bottom corner of the graph due to their characteristics of a rather long duration of 2-4 days and slightly low intensities. Only the event of 2019, 2008 and 1953 are dispersed on the other side of the graph where the duration is around or less than a day.

Considering the thresholds proposed by Guzzetti, all the events are correctly settled above. No significant differences are seen among the general one (b), the curve valid for mid-latitude climate (c) and the one valid for highlands climate (d). Peruccacci (e) and Crosta-Frattini (h) pose intermediately between the regional threshold of Guzzetti and the local ones

proposed by Cancelli-Nova (f) and Ceriani (g). It seems that Guzzetti, Peruccacci and Crosta-Frattini may overpredict critical events because they are positioned rather low, especially for short duration ones.

The thresholds proposed by Caine (a), Cancelli-Nova (f) and Ceriani (g) are placed above the previous ones. The Ceriani curve seems to fit very well the data, positioning only the 1966 event slightly below the curve and the 1953 and 1960 close to the curve. Also, Cancelli Nova works rather well posing only 1953 below the threshold. These results were expected because both (g) and (f) threshold were calibrated using a local dataset, respectively up to 1985 and 1994. Conversely, the Caine threshold seems to work worst rather than the previous leading to underprediction: 1953, 1960 and the 1966 events are not identified as critical and appear below the curve. Moreover, the 1997 and 2000 are settled borderline on the curve.

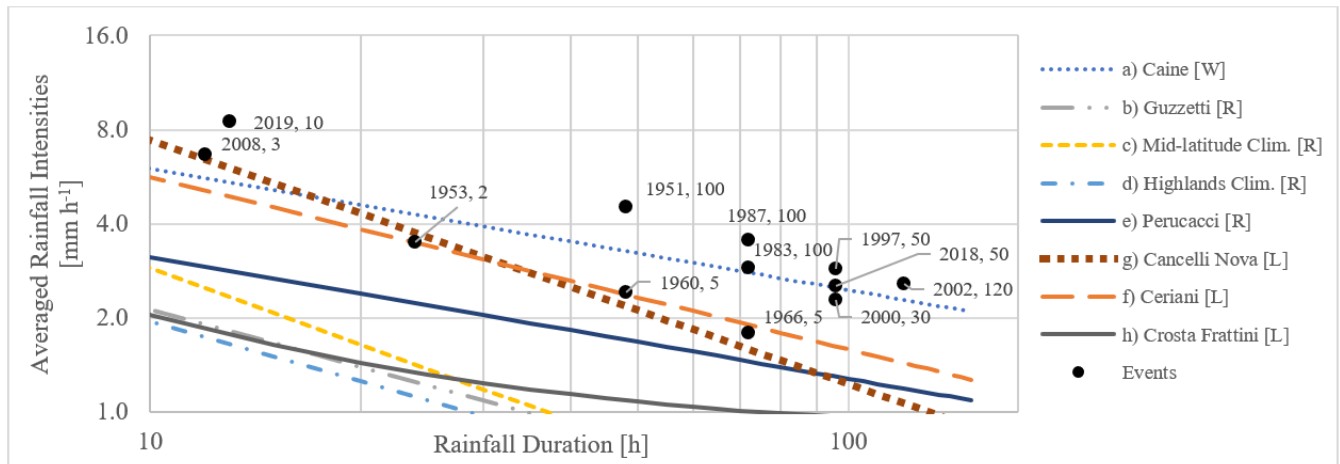

Figure 5: Intensity-Duration thresholds for considered events and relative rainfall RP.

The threshold curves analysed have divided our events into critical and non-critical ones but no further information on their magnitude was retrieved yet. Some authors have shown that a measure of magnitude may be established considering the relative distances between the I-D points and the threshold curve. According to (Crosta and Frattini, 2001; Gao et al., 2018; Iida, 2004; Rosso et al., 2006), a beam of rainfall I-D curves can be elaborated including their dependence on RP. For the same area, rainfall events with higher RP should be statistically located much more distant from the threshold lines, but this fact strongly depends on the reference curve considered as the lower bound. In our study, local thresholds of Ceriani and Cancelli-Nova have demonstrated to best fitting the dataset avoiding under- and over predictions. Moreover, they are delimited by 1953, 1960, 1966 and 2008 events which exhibit the lowest RPs comprised between 2-5 years. Taking these curves as a reference we can appreciate that other critical events showing higher RPs are also located at more distance from these curves. This represents a confirmation of what found in the literature, but, in our opinion, the magnitude assessment looking simply at relative threshold distance seems rather approximate. In fact, the RP estimation depends not only on rainfall I-D values but also on parameters of GEV that takes into account the spatial variability of local precipitation statistics (De Michele et al., 2005). In those cases where rainfall intensity and duration are fixed, changing the GEV parameters also the RP may vary even though the relative distance from the curve is the same. In our dataset, we have

encountered this fact two times comparing 1983 and 1987 events, and 1997 and 2018 events that respectively exhibit the same RP with the same duration but a different relative distance from the curves. As a result, these distances could be used as a proxy of the magnitude only for rainfall analysis carried out at the same location where the GEV parameters remain constant, confirming what suggested by other authors. In our case study, this condition was not satisfied because the GEV parameters were not constant in space.

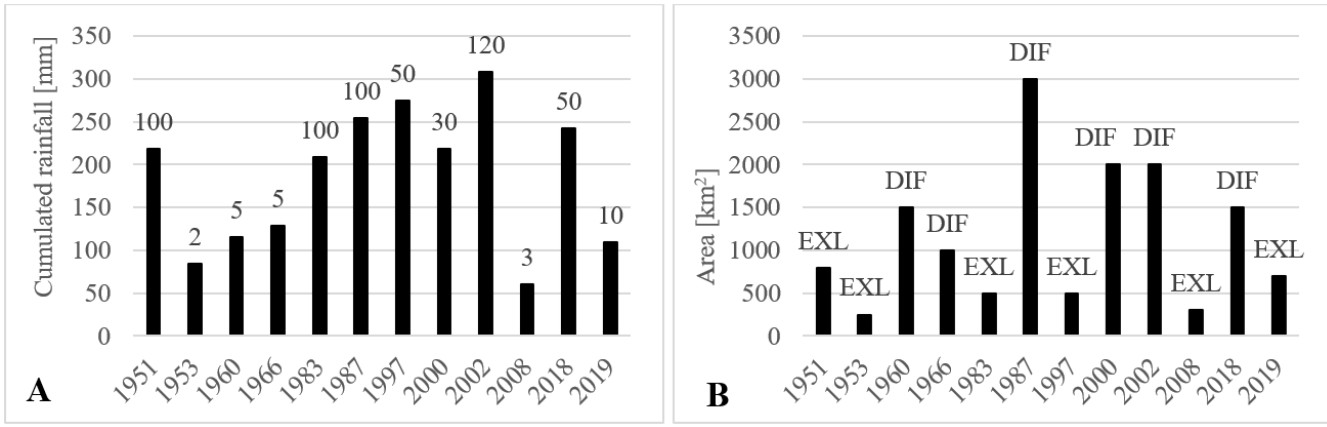

Figure 6: Cumulated rainfall and RP of triggering events A) and the Area affected by geo-hydrological issues B).

Looking at Figure 6.A, the Sondrio Province has experienced at least four exceptional rainfall events with a return period equal to or higher than 100 years: 1951, 1983, 1987 and 2002. From RP analysis, they were ranked with the same intensity but among them, 1987 has been recorded historically as the most catastrophic one that affected the area in the second half of the XX century. This apparent contradiction has a possible explanation if we also include the information about the spatial extension of the triggers, as reported in Figure 6.B, which is a property strictly related to the nature of the rainfall event (Corominas et al., 2014; Gao et al., 2018). This parameter is not explicitly considered in RP evaluation. As an example, we can compare the 1983 and 1987 events. If only the RP is considered, 1983 intensity is equal to 1987, but considering the spatial distribution, the 1983 event affected only a limited area while 1987 spread across the entire province. For this reason, if we are interested in determining the magnitude of meteorological triggers, 1987 should be intended more critical rather than 1983. In this regard, the RP information could be misleading.

According to (Corominas et al., 2014; Guzzetti et al., 2005) and following the methodology proposed in Eq. (2 to 3) we have moved further considering both RP and AA for determining the trigger's hazard and magnitude. First of all, the FMCs have been established, allowing us to define the probability of spatial and time occurrence as a function of parameters AA in Figure 7.A and RP in Figure 7.B. Secondly, AA has been plotted against the RP in Figure 8.A and was observed their low statistical correlation. Then, considering Eq. (3.a), the trigger's hazard has been defined and reported in Figure 8.B. We can notice that the trigger's hazard is higher when higher are the probabilities of spatial and temporal occurrence. In particular, 1953, 2018 and 2019 represent the most hazardous events with lower RP and AA. On the other hand, 1987 and 2002 represent the lowest hazardous events because, from a probabilistic viewpoint, they exhibit both the highest return period

and extension. Applying the Eq. (3.c) the trigger's hazard has been translated into the magnitude index MI, normalized in respect to its maximum and shown in Figure 8.B. We can notice that the MI has highlighted 1987 and 2002 as the most severe events. On the other hand, 1953 and 2008 were depicted with the lowest magnitudes. An intermediate magnitude ranking was assessed for 1951, 1983, 1997, 2000 and 2018 events confirming historical evidence.

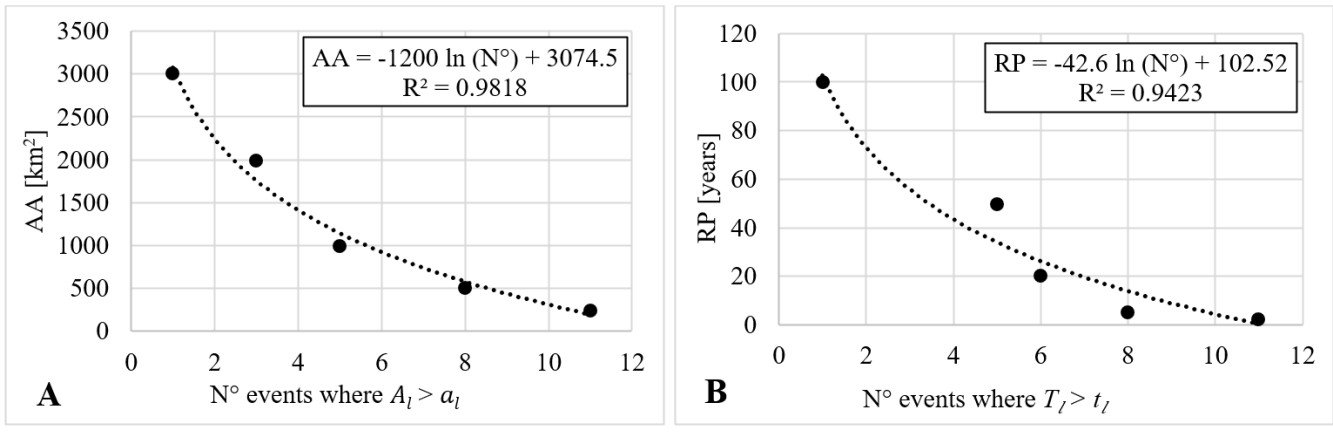


Figure 7: Frequency-magnitude relationship for A) Area Affected (AA) parameter and B) Return Period (RP) parameter.

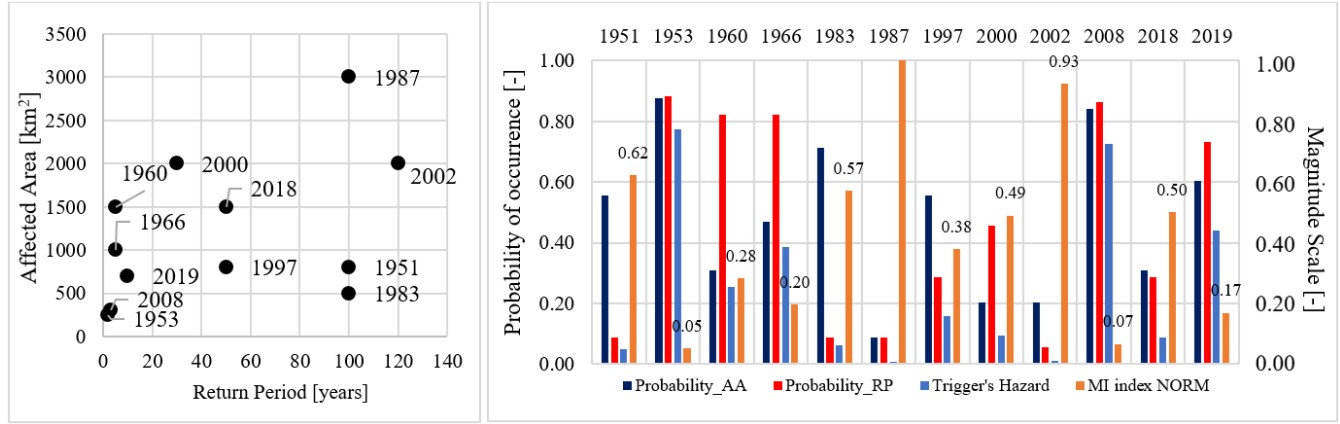

Figure 8: A) Correlation between RP and AA parameters, and B) determination the probability of occurrence of AA, RP and the Trigger's Hazard for dataset events

**4.3 Approach 2: ECs intensity analysis and SLPT index**

In the second approach, we applied the NCM described in Eq. (4). Using the NCEP data, atmospheric pressure gradients, wind velocities, and air masses advection through the Alpine region the model components in the Eq. (4.a2), (4.b2), (4.c2) and (4.d3) were studied.

For determining the $T_1$ term (Eq. 4.a2, upper layer divergence), the geostrophic wind velocities were estimated. Geostrophic
wind is the theoretical wind that would result from an exact balance between the Coriolis force and the pressure gradient force. It represents a first approximation of the general circulation of the air masses at a regional scale. Intense geostrophic

velocities are generally associated with strong EC structures (Andrews, 2010; Martin, 2006; Stull, 2017). As reported in Figure 9.A, geostrophic velocities were higher for 1983, 1987, 2000 and 2002, a sub-group of the most intense events of our dataset. Upwind velocities in Figure 9.B are also correlated with the presence of sustained geostrophic winds. Again 1987,
2002 and now 2018 have shown the highest values of the entire dataset.

For determining the $T_2$ and $T_3$ terms (Eq. (4.b2) boundary layer pumping and Eq. (4.c2) advection), the air masses evolution paths were examined. Figure 9.C shows the short distance $\Delta s_2$ between the low pressure (L) and the Sondrio Province. We can notice that the relative position of ECs does not vary too much, 1183 km on average. This represents a characteristic of the ECs structures that tends to evolve across the Mediterranean and the Alpine area similarly. Nevertheless, some seasonal
changes can be appreciated by looking at the advection path followed by the low-pressure centre (L). The larger part of the autumnal events exhibits a meridian motion of the low pressure from the northern part of Europe (Northern Sea) to the southern part, entering the Mediterranean Sea and moving eastward following Rossby waves track (Rotunno and Houze, 2007; Stull, 2017). This is the case of 1960, 1966, 2000, 2002 and 2018 events that occurred between September and November. Summer events of 1951, 1953, 1987, 1997 and 2019 exhibit a low-pressure tracking path that did not cross the
Alps mountain range. This fact can be explained by considering that Rossby waves are in general shifted northward during the summer period (Grazzini and Vitart, 2015; Martin, 2006). This reflects on the events that affect the southern side of the alpine region which are more rapid, less persistent, locally intense but not well organized such as the typical autumnal EC.

The $T_4$ term is represented by a linear function of the daily rainfall rates RR considered in the precipitation analysis. In the formulation adopted we made strong assumptions to yield the problem more tractable. This is the only component that
depends on the accurate estimation of the ground-based rainfall data.

After calculating the intermediate components $T_1, T_2, T_3$ and $T_4$ terms, the Sea-Level Pressure Tendency index (SLPT) of Eq. (4) has been determined, Figure 9.D. Firstly, we can notice that all these ECs have been characterized by explosive cyclogenesis. This definition applies when an extratropical cyclone exhibits a low pressure deepening of 24 hPa in 24 h, which corresponds to an average rate of 1 hPa h$^{-1}$ (Sanders and Gyakum, 1980). Looking at Figure 9.D, the SLPT index
shows a range comprised between the – 2.64 hPa h$^{-1}$, recorded for the 1953 event and -4.89 hPa h$^{-1}$ recorded for 1987. The latter and 2002 (-4.67 hPa $^{-1}$) are reported to have been the EC structures with the highest intensity that affected the Northern Lombardy area. An average value of the SLPT index is reported around -3.67 ± 0.63 kPa h$^{-1}$ that is compatible with the ECs structures shown by NCEP maps.

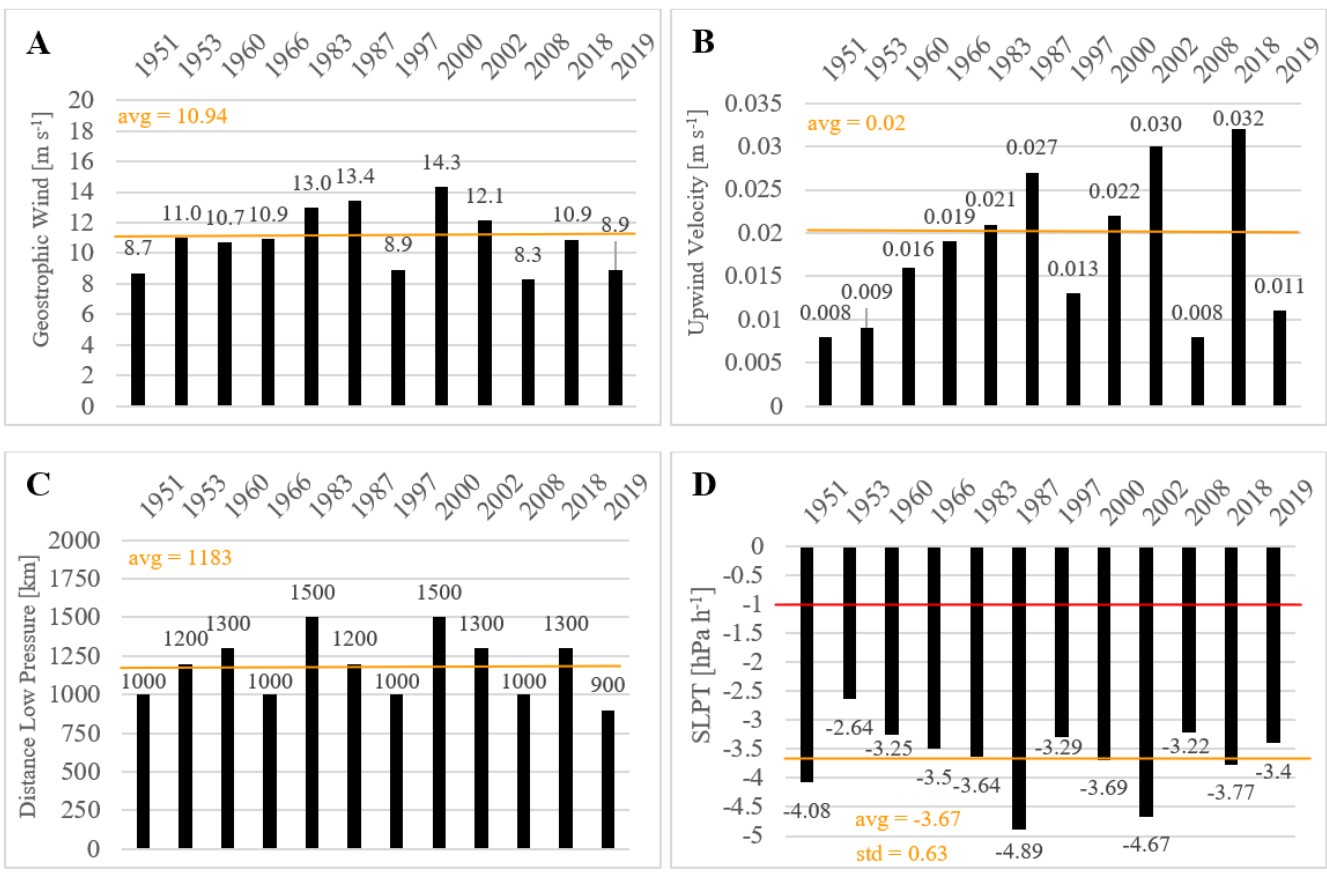

Figure 9: A) Upwind velocity and B) geostrophic wind velocity calculated for $T_1$ term and C) $\Delta s_2$ considered for $T_2$ and $T_3$ terms. D) The Sea-Level Pressure Tendency Index (SLPT) for the event analysed is compute. Orange lines represent the averages across the dataset while the red line indicates the threshold of explosive cyclogenesis (1 hPa h⁻¹).

## 4.4 Comparison between MI and SLPT indexes

The two methodologies proposed for the trigger's magnitude assessment are now compared. The two indexes MI and SLPT have been firstly normalized in respect to their maximum and then shown in Figure 10.A. We can observe that it is rather clear how the two indexes give a similar magnitude rank for the events examined in our dataset. Looking at bias errors, the mean absolute error (MAE) is computed around 7 % and the root mean square error (RMSE) is also about 10.3 %. The highest absolute error values were addressed by the 2008 and 2019 events. Moreover, we can show that two indexes are in accordance identifying 1987 as the episode with the highest magnitude, followed by 2002 and 1951. The lowest ranking scores are established for the 1953 event and 2008 that were already spotted by I-D analysis as borderline for Cancelli-Nova and Ceriani thresholds. In the middle, we found 1960, 1966, 1983, 1997, 2000, 2018 and 2019, which were depicted also by historical chronicles as rather intense but not catastrophic for the Sondrio Province. In Figure 10.B the MI index and the SLPT index have been plotted against each other. From Figure 10.B can be appreciated that the points lay on the diagonal and the correlation index $R^2$ is about 0.88, which is rather high and near to 1.

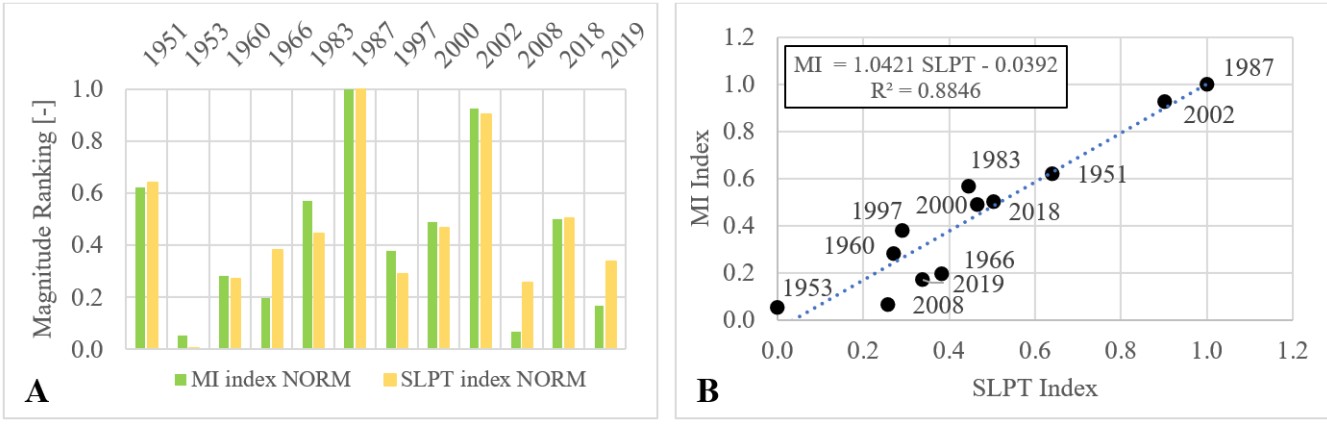

Figure 10: Comparison between MI index and SLPT index, normalized. The events recorded in 1951, 1987, 2002 and 2018 has highlighted with the highest magnitude while 1953 and 2008 events have the lowest values.

## 5. Discussion

Considering the results obtained, we discuss here the questions that aimed at our study. The first was: "Are the I-D thresholds and the RP evaluation enough for a complete description of meteorological triggering factors?" The I-D thresholds are typically used for geo-hydrological risk assessment but some uncertainties about their reliability have risen around two aspects: the choice of the best-fitted threshold and the threshold's dependency on the RP parameter.

Regarding the first aspect, the thresholds can distinguish critical or non-critical events giving only a binary outcome of the event classification. Shifting up – down the curve or changing the curve, the same event can be detected respectively as a false negative or a false positive and this fact may lead to a prediction error. In the specific case of our dataset, Guzzetti, Perrucacci and Crosta-Frattini curves seem to overpredict these events while Caine was found to underpredict them. On the other hand, Cancelli-Nova and Ceriani have demonstrated more suitable for interpreting our dataset. In this regard, the local thresholds seem to be more accurate rather than the regional ones, but uncertainties remain about their correct application and interpretation. In fact, some recent studies have suggested that further investigation around their parameter's definition are required to improve detection performances. According to several authors (Bogaard and Greco, 2018; Kim et al., 2020; Lazzari et al., 2018) the threshold may exhibit dynamic behaviour, shifting up-down considering the soil moisture and the antecedent cumulated rainfall especially for short duration events. This important condition has been normally neglected in the past definition of the thresholds, treating all the triggering events as uniform from the statistical point of view. Therefore, a wise disaggregation of these events in the function of antecedent conditions should be applied for creating a new threshold set that highlight the sensibility to those variables. In our opinion, this may help to improve further the performance of I-D methodology especially for locally based thresholds under the reasonable hypothesis of a uniform spatial susceptibility of the territory. On the other hand, for the regional ones, we think that the improvements would be less effective because also other

factors related to the more heterogeneous area, such as morphological or geological predisposing causes, may play a more important role (Peruccacci et al., 2017). Including the RP in the threshold analysis can be useful to determine a preliminary magnitude ranking. Even though higher RPs are generally founded at a higher distance from the curve, the relative distance among I-D point and the reference threshold cannot be always considered as a proxy of the event magnitude. According to (Gao et al., 2018) this assumption has been reported not so strong, and this was confirmed also in our reanalysis study. A

possible explanation can be found in the way the RPs are estimated. In principle, this interpretation of the trigger's magnitude is still valid only at a very local scale but cannot be adopted in our study since the GEV parameters used in RPs have changed in each rainfall episodes. Our results have highlighted this fact two times showing different point-threshold distances with respect to the same RP values. In this perspective, climate change will pose some challenges about the GEV updating for the future, considering that no stationary processes could affect the statistical distribution of critical

precipitation (Albano et al., 2017b; Gariano and Guzzetti, 2016). This may add further uncertainties to this interpretation that considers only I-D thresholds and RPs for event magnitude estimation.

These two important observations represent a critical point in the I-D threshold methodology that has driven us to ask: "Is RP a good predictor of the magnitude?" Typically, the magnitude of a rainfall episode is described by the RP value, but this information is evaluated only from a time perspective. Taking inspiration from the landslide hazard definition proposed by

(Guzzetti et al., 2005) we defined a new magnitude index, MI, that was also representative of the "triggers energy". In the definition of MI, we have included the information about the trigger's spatial distribution AA. This choice was aimed at the lack of precise data about the landslide volumes, extensions, or numbers, which are quantities considered for assessing an event magnitude scale (Malamud et al., 2004). The AA parameter can be interpreted as another proxy of the trigger's magnitude because indirectly it can describe the nature of the rainfall phenomena, distinguishing between a heavy

thunderstorm, localized, in respect to persistent rain, more diffused. As shown by our results, RP and AA were uncorrelated so both were considered for the assessment of the magnitude index MI. The MI index was estimated in our study with post-event information but theoretically the index can be evaluated using weather forecasting, looking at expected rainfall rates and amounts across different areas. In this regard, Local Area Meteorological Models (LAMs) can be used to estimate the MI index some hours in advance of the event. In our opinion, this represents one of the main advantages of using MI

because, in respect to the other magnitude indexes that requires precise information about the "post-failure" effects (number of triggered landslides or peak discharge), the MI can be established using again only meteorological information, much like the SLPT index that we propose further.

As a matter of fact, we have implicitly answered the third question proposed: "Can rainfall analysis be improved considering also other meteorological variables that are related to the trigger's magnitude?" The assessment of the MI index has

highlighted that the very local information about precipitation is not exhaustive, and spatial distribution of the rainfall is also needed to better comprehend the differences among the events. Moreover, if we are interested in the accurate trigger's description, looking only at the "final product" of a more complex meteorological process maybe not enough (Monitoring European climate using surface observations; Rotunno and Houze, 2007; Stull, 2017). This is particularly true in mountain

areas where the territory enhances the heterogeneity of the rainfall field (Abbate et al., 2021). For these reasons, other meteorological variables should be taken into account and included in the analysis. In our study, to pursue this goal we moved from a local perspective to a more regional one. This is crucial because it permits to better describe the different precipitation type that may influence the occurrence of geo-hydrological failures (Corominas et al., 2014; Guzzetti et al., 2007). As an example, an intense thunderstorm during summertime could trigger few shallow landslides or debris over a limited area (Abbate et al., 2021; Montrasio, 2000) in respect to a persistent orographic rainfall that could affect an entire region, trigger diffuse terrain instabilities and reactivate also deep-seated landslides (Longoni et al., 2011; Rotunno and Houze, 2007; Tropeano, 1997). In this regard, the local rain gauges series have been integrated with the NCEP reanalysis maps data and the SLPT index was evaluated applying the theory of the Norwegian Cyclone Model. The implementation of this methodology has represented an innovative way to gain a comprehensive meteorological description of the rainfall triggers. In fact, in the NCM model, the ground-based rainfall series represent only one term ($T_4$) that is involved in the EC intensification. The former depends also on other processes: the upper layer divergence ($T_1$), boundary layer pumping ($T_2$) and low-pressure advection ($T_3$). This additional information has been addressed to play an important role in EC evolution and helped us on better differentiate critical events characteristics.

The SLPT index formulation requires several data about triggers. These can be retrieved easily by looking at a reanalysis database such as the NCEP reanalysis maps. However, NCEP maps interpretation is rather useful only for past events. Nowadays LAMs are much more suitable for interpreting the mechanism of EC through a complex orographical area like the Alps (Ralph et al., 2004; Rotunno and Houze, 2007). In this regard, the NCM model is still valid but the processes involved can be interpreted at a higher detail level with LAMs, avoiding some of the hypothesis required by NCM. The evaluation of the SLPT index should be intended as propaedeutic to further analysis and it cannot be adopted in every situation. As we have foreseen from results, concerning I-D thresholds methodology, the SLPT estimation requires moving from a very local perspective to a regional scale. This operation makes sense if the investigated area is rather extended for excluding very site-specific chain effects that can be triggered by isolated rainfall episodes, such as thunderstorm cells. Another important limitation on the applicability of the SLPT index regards the presence of a recognizable EC's structure from meteorological maps. In fact, for weak EC's, the estimation of the trigger's magnitude may bring larger errors. In our study, this fact was experienced for the cases of 1953, 2008 and 2019 and was confirmed through visual inspection of NCEP maps. In these situations, the rainfall analysis should be restricted to a more local domain trying to include also LAMs outputs, radiosonde, and satellite data (Abbate et al., 2021) and the application of the MI index could be much more appropriated for the magnitude estimation.

As a result of our study, we have compared the two MI and SLPT indexes to assess the magnitude of critical events. Even if they come from different theories, MI is based on frequency-magnitude theory and SLPT is has a physical meaning in the meteorology field, appears clear how they are in accordance depicting the same critical events with the highest magnitudes. This outcome has found confirmation in the qualitative information we retrieved in the historical database. These results have demonstrated that exists a strong cause-effect relationship among the strength of EC developed at a regional scale in

respect to the effects recorded on a local scale, especially for strong events. For the dataset examined, the SLPT comparison with the MI index was rather encouraging, $R^2 = 0.88$, and the additional information retrieved from NECP maps has sharply improved the rainfall reanalysis completeness. In our opinion, both proposed indexes are useful instruments for describing the magnitude of the rainfall-induced events, overcoming the uncertainties of the I-D threshold methodology.

## 6 Conclusions

This study presents an extended reanalysis of the meteorological triggering factors that have caused in the past several geo-hydrological issues in the alpine mountain territory of the Sondrio Province, Northern Lombardy, Italy. Excluding the geomorphological predisposing causes of the area, the attention was pointed out to the characteristics of the rainfall. The main goal of our study was to assign a quantitative magnitude ranking to the meteorological trigger, following two approaches.

In the first one, the I-D threshold curve analysis was considered to identify critical rainfall events. We have demonstrated that the events fit some I-D thresholds, in particular the local thresholds of Cancelli-Nova and Ceriani, and that the distance from the curve does not necessarily mean that an event has a higher RP. For this reason, to assign a magnitude to each of the events, we proposed the MI index, which integrates the return period and the spatial extent of the event. The MI index was determined analytically starting from the frequency-magnitude theory, under the hypothesis that the event's magnitude was also a function of the spatial distribution of the trigger, described by the parameter AA. In the second approach, the trigger's analysis was conducted from a simply meteorological viewpoint evaluating the strength of extratropical cyclone structure through the NCM model. Using the information of NCEP reanalysis maps the SLPT index was determined and interpreted as another trigger's magnitude index, much like the MI.

The two indexes have been compared showing good accordance in the assessment of a magnitude ranking for the studied events. The SLPT index has confirmed the important relationship between the EC's intensity at a regional scale and the correspondent trigger's magnitude recorded locally, described by the MI. The two indexes are based on meteorological data, therefore, may found an application in the now-casting meteorology field. This could represent an important advancement, especially for the early warning systems adopted by municipalities for geo-hydrological risks mitigation.

In view of the future climate change that, with high confidence (Faggian, 2015), will affect the Mediterranean and the Alpine environment, extreme meteorological events are supposed to increase (Ciervo et al., 2017; Gariano and Guzzetti, 2016; Moreiras et al., 2018) and also geo-hydrological hazards may rise in frequency. Our study moves in this direction, trying to extend the interpretation of rainfall triggering factors through a more meteorological perspective.

*Code and data availability:* All the data reported in this paper are freely consultable on Internet websites. In particular, reanalysis weather maps are freely downloadable from Meteociel Website (MeteoCiel, 2020), IFFI and AVI database are freely consultable and downloadable from (Sistema Informativo sulle Catastrofi idrogeologiche; Inventario Fenomeni

Franosi), and rain gauges data are extracted from local Environmental Agency (ARPA Lombardia, 2020). The model applied in this work is also freely consultable and downloadable from (Stull, 2017).

*Author Contribution:* Andrea Abbate and Laura Longoni conceptualized the study. Andrea Abbate carried out the formal analysis and wrote the manuscript with contributions from all co-authors. Laura Longoni and Monica Papini supervised the
research and all the authors reviewed & edited the manuscript.

*Competing Interests:* The authors declare that they have no conflict of interest.

*Acknowledgements:* The authors acknowledge the support provided by Fondazione CARIPLO through funding the project MHYCONOS, grant number 2017-0737 and to the "Geoinformatics and Earth Observation for Landslide Monitoring"
project financed by "Ministero degli affari esteri e la cooperazione internazionale", in cooperation with Hanoi University of Natural Resources and Environment, Vietnam.

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

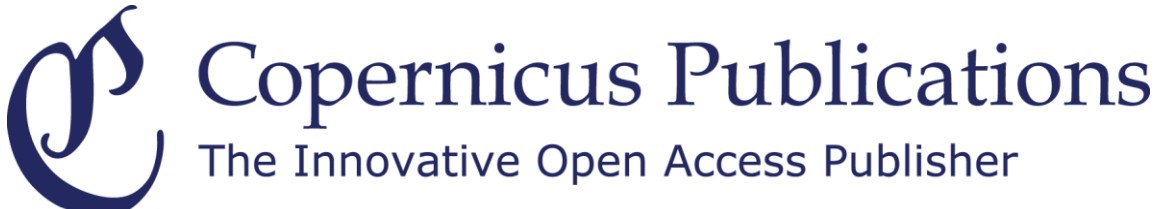

**Fig. 11: The logo of Copernicus Publications.**


