# Peer review of "Analysis of meteorological parameters triggering rainfall induced landslide: a review of 70 years in Valtellina"

_Natural Hazards and Earth System Sciences, 2020_

## Referee Comment (RC1) · Anonymous Referee #1 · 16 Jun 2020

Manuscript Number: https://doi.org/10.5194/nhess-2020-118

Title: Meteorology triggering factors analysis for rainfall induced hydrogeological events in alpine region

Overview and general comments:

Authors studies the past meteorological (mainly rainfall) conditions that lead to certain hydrogeological events. They applied a systematic analysis, and fairly self-criticize their results. The study is relevant for particularly landslide researchers that use rainfall I-D curves. Although the technical frame work of the manuscript seems sound and robust, the presentation of the work is rather poor in the current state. Overall, I got

the impression that this manuscript is not properly edited before submission. There are several tiny mistakes in the text, which makes it hard to follow. With proper editing the manuscript will get considerably shorter and understandable. This manuscript deserves publication, after considerable changes in its current presentation.

Major comments:

Authors evaluate/discuss the rainfall I-D curves only based on "false negatives", but ignore "false positives". A certain rainfall might remain above the I-D curve of Guzzetti et al. (2007) or Mid-latitude Climate without triggering any landslide as well.

The authors avoid discussing their results on a broader scale beyond their study area. They rather give an event summary in the analyses. They should extend their discussion and explain what do we learn from this study that is valid also for other high relief areas?

Minor comments:

There are wrong use of words and tenses throughout the manuscript. It should be proof read. Authors prefer to use passive voice; I believe this practice is not recommended anymore.

Excessive use of connecting statements, such as "on the other hand", "conversely", "Therefore", "In conclusion". Please omit the ones that are not absolutely necessary.

Title can be a bit punchier.

Abstract

It is long and it lacks providing a motivation and a take home message. For example, first two sentences of the abstract sounds similar. There are also a few sentences without any information, e.g. "The results obtained from the application of the two methodologies have been discussed.". It also does not involve any clear take home message at the end, as expected.

Introduction The introduction is really hard to follow, it does not develop logically leading to the research questions of - Are these approaches sufficient for a complete description of triggering factors? - Can rainfall analysis be improved considering also other meteorological variables, which could better describe the rainfall events and the linked consequences? Why "other meteorological variables" are not mentioned before? Which approaches could be an alternative? These questions are not returned at the discussion one more time.

A topic in a paragraph is returned after a few paragraphs again that confuses the reader. There are a few references, e.g. Guzzetti; Rosi; Gao, that appear frequently in different paragraphs. Authors should review the global literature carefully and reformulate the introduction leading to the research questions also beyond their study area.

Line 47: "However, for shallow landslides. . ." I can't understand this sentence.

Line 52: "These thresholds data are calibrated looking at the past events occurred in the area and directly correlated with the nearest rain gauge measures (Rappelli, 2008)." Rainfall intensity might vary considerably during an event, gauge data might miss this variation. We see this especially when rainfall radar estimates are compared with the gauge observations. Authors should also mention this effect in the same paragraph.

Line 58: ". . .rainfall thresholds have been widely used in different parts of the world.", but the authors refer usually to the studies from Italy, if I am not wrong.

Line 74-82: Similar statements are mentioned in the Abstract as well as early in the Introduction, please refrain using same statements again and again.

Data, Methods and Models Authors should consider using dedicated chapters as "Data" and "Methods".

The method part explains how the computation works but lacks information about the meaning of the results for the current study.

Line 96: "...estimated in 2 billion of euros." reference missing.

Line 98: "...glacier melting increased by high-altitude summer temperatures." reference missing.

Line 143: "...extratropical cyclone structures" Different type of rainfalls that effect the regions might be important for the entire study. Authors should consider providing more info about these effect in the introduction.

Line 146: "...the traditional rainfall approach and the meteorological reanalysis approach." They are not mentioned before.

Line 154: "...by several authors.' Who are they and how they mention it?

Line 187: "...extratropical cyclone (EC), as described in Figure 2" I am not sure whether I can see the message in the figure.

Results and Discussion

It is hard to follow what is the new result and what is the discussion point. Please consider using a dedicated section for each, "Results", "Discussion".

Line 236: "...rain rate ðİŘij..." I guess rain intensity Line 255: "...possible indicator of the magnitude of the hydrogeological events..." ...of...of... Line 262: "...does not permit...", do you mean "hinders", I recommend authors to use either negative prefixes and suffixes, or negative verbs directly instead of negative verb conjugation throughout the manuscript. For example, in line 281: "...it does not exist a unique method for the magnitude assessment..." → "...a unique method lacks that assess the magnitude..."

Conclusion

The section is dedicated to summarize the applied analyses, I could not really find a clear take home message that evolved from the results and discussion.

Line 384: What are those "hydrogeological issues"? After reading the entire

manuscript, I am still not sure about this term? Are the authors refer to hydrogeological evets, such as rainfall induced landslides? Or are there other events?

Tables:

Table 1: The caption is repeating the column titles of the table.

Figures: There are several bar plots, which does not provide so much information. They may be fit for the appendix, but they are poor for the main body of the manuscript. Authors should consider re-generating figures that provide clear messages. They could consider combining several of the figures in a more creative manner.

Figure 1: X and Y labels and ticks are missing. Fonts are larger than in the main text.

Figure 2: What is the purpose of the arrows?

Figure 4: What is mid-latitude-and highlands climate?

Figure 6 and figure 7: I cannot understand the message of these two, especially the figure 7 shows nearly constant Geostrophic velocity ($\sim$42+-5 km/h)

Figure 8: Location of the Alps are different in each subplot; Lat Lon data is missing the numbers on the isolines are too small to read.

---

## Author Comment (AC1) · 26 Jun 2020

We would like to thank the reviewer for the time and effort he/she put into thoroughly reviewing this manuscript. We believe that the comments are constructive and would lead the considerable improvement of our work. During the revision we will carefully address the suggestions raised by the reviewer along with any other recommendation potentially emerging during the window for open discussion.

Your Sincerely

The Authors

---

## Short Comment (SC1) · 26 Nov 2020

Dear authors,

I think your research is interesting and valuable for the hydro-geological research community especially those who are analyzing events in the Alpine region. However I do not see that your manuscript being formulated explicitly enough to emphasize such value of forensic event analysis, despite the lack of novelty in the methodology. I would suggest to reformulate the Introduction (and abstract) and the conclusion to include more discussion on your analysis limitations as well and the outlook to further improve such kind of forensic analysis.

Some specific comments: figure 5, label of event (DATA), what is the value after comma? Also I think DATA is better called 'observation'.

table 1, estimated affected area (km2) is this the affected area based on the disaster extent or the rainfall extent? In this case rainfall spatial extent may be much smaller than the disaster extent and the other way around, as surface processes and effective rainfall intensity may be pose as a dominant role.

For discussion: As some of the rainfall events may be small in spatial extent but intense within sub-daily duration, can this characteristics be captured by the coarse resolution reanalysis, and the possibly sparse rainfall gauges data? Follow up to this, I think it would also be interesting to look at the antecedent soil moisture (e.g. implied by antecedent precipitation) prior to the events, as the saturation level of the soil easily trigger intense surface runoff in spite of the moderate rainfall, as often seen in flash floods. That said, I also think the paper does not sufficiently mention the contributing factors of hydrological and surface processes, as the lack of such can also mislead the use of 'meteorological only' analysis.

Good luck and keep up the good work.

---

## Referee Comment (RC2) · Anonymous Referee #2 · 24 Dec 2020

The manuscript deals with an interesting attempt to strengthen the estimation of extreme hydrological conditions, which led to landslide and flash flood phenomena in the Sondrio Province (northern Italy) in the period 1951-2019, by linking them to estimations of magnitude and return period (temporal probability) as well as meteorological conditions occurred at the continental scale. In the opinion of this reviewer, such a methodological effort is certainly to be appreciated because tending to reduce uncertainties of a single approach, such as empirical rainfall thresholds for shallow landslide triggering and hydrological probabilistic models. Specifically, starting from an inventory of principal landslide and flash flood phenomena collected for the study area, authors analyzed triggering hydrological factors by three methods. The first is based on

the comparison of intensity/duration of rainfall events, which led to landslide and flash flood phenomena, to empirical rainfall thresholds for shallow landslides known by the literature for the region studied and worldwide. The second is the estimation of return periods for triggering rainfall conditions by a known regional probabilistic model (De Michele et al., 2005). The third is the analysis of meteorological conditions which lead to extreme rainfall events, based on data taken from the National Centres of Environmental Prediction (NCEP) (Kalnay et al., 1996; MeteoCiel, 2020; NOAA, 2020). Notwithstanding the challenging and innovative premises, the manuscript presents several conceptual points of weakness which are described below in form of both general and specific comments.

GENERAL COMMENTS 1) Rainfall thresholds considered, known by the literature, are related to shallow landslides, instead Authors compare to them also deep-seated phenomena (1987 deep-seated Val Pola landslide) and flash floods. Therefore, the comparison appears too heterogeneous and would need a motivation. 2) The procedure used for assessing the magnitude ranking (Fig. 6), by taking into account of both return period and areal extent of meteorological phenomena, appears questionable for being based on the mean of normalized value. By a conceptual point of view, considering the mean value is allowed for the same variable, not for different variables. Very likely, the normalized product of return period and areal extent of meteorological phenomena would incorporate more consistently the magnitude at local scale (return period) and the areal extension. 3) Parts of the paragraph 3.2.1 regards general methodological aspects and preceding knowledge, therefore they appear more suitable for the methods section rather than the results one. 4) Nothing is given about the regional probabilistic model (De Michele et al., 2005), which has been considered to the estimation of return periods. In the opinion of this reviewer, it's an important point to explain aspects related to the regional probabilistic model adopted. 5) Authors should motivate with a greater emphasis the possible applications of their findings in the field of landslide and flood hazard assessment as well as early warning systems. 6) English language should be revised.

SPECIFIC COMMENTS

The adjective "hydrogeological", used extensively throughout the text, appears not suitable to indicate landslide and flood phenomena. It is recommended to substitute this term.

Block 20. The sentence is not clear and it should be rewritten.

Block 30. The use of the term "back analysis" is questionable because it is commonly used in the geotechnics field for inverting a slope stability analysis and estimating shear strength of geological materials involved in landsliding. In this case the analyses carried out are just re-examinations of past landslide and flash flood events.

Block 40. Substitute "deep landslide" with "deep-seated landslide".

Block 80. Substitute "will be" with "is".

Block 90. "Old debris" is not a geological term. Maybe, just debris could be better, otherwise the age should be indicated more clearly (e.g. Pleistocene).

Table 1: A column indicating the mean intensity should be considered. The definition of Extremely Localized (EXTL) and Diffuse (DIF) could be substituted with Localized (L) and Areal (A).

Blocks 250-255 and Fig. 4. To consider the vertical distance between the curve and the critical event point appears conceptually incorrect due the possibility that the curve itself (I/D rainfall thresholds) indicates points with different return period. Authors should verify and discuss this point.

---

## Author Comment (AC3) · 29 Dec 2020

REPLY TO REFEREE 1

Title: Meteorology triggering factors analysis for rainfall induced hydrogeological events in alpine region

We are kindly grateful for your accurate revision of our work. We have really appreciated your hints and suggestions and in this brief reply we are going to discuss them.

Overview and general comments: Authors studies the past meteorological (mainly rainfall) conditions that lead to certain hydrogeological events. They applied a systematic

analysis, and fairly self-criticize their results. The study is relevant for particularly landslide researchers that use rainfall I-D curves. Although the technical framework of the manuscript seems sound and robust, the presentation of the work is rather poor in the current state. Overall, I got the impression that this manuscript is not properly edited before submission. There are several tiny mistakes in the text, which makes it hard to follow. With proper editing the manuscript will get considerably shorter and understandable. This manuscript deserves publication, after considerable changes in its current presentation.

We are aware that the work is not properly ready for a direct submission because several topics we have analyzed should be more integrated.

MAJOR COMMENTS Authors evaluate/discuss the rainfall I-D curves only based on "false negatives", but ignore "false positives". A certain rainfall might remain above the I-D curve of Guzzetti et al. (2007) or Mid-latitude Climate without triggering any landslide as well. The authors avoid discussing their results on a broader scale beyond their study area. They rather give an event summary in the analyses. They should extend their discussion and explain what do we learn from this study that is valid also for other high relief areas?

Our goal and the scope of the paper was to extend the Rainfall analysis. Commonly, rainfall is considered as a precursor of the shallow movements of terrain but due its spatial variability on complex territory can lead to false-negative, as we have assessed in the paper or false-positive, depending on the I-D curve considered. In this we agree that a broad discussion about I-D method uncertainties should be extended: site specific curves, poor rainfall data also in the Alps that is the most monitored range all over the world, radar failures in complex terrain etc. that may help or not to correct the rainfall intensity estimation. These facts bring to an approximate representation of the rainfall intensity and led to wrong interpretation of triggering events using this approach. So, it is useful to move to other approaches to estimate the intensity of the triggering events considering also other meteorological variables that are correlated with but can reduce

the uncertainties in rainfalls analysis.

MINOR COMMENTS There are wrong use of words and tenses throughout the manuscript. It should be proof read. Authors prefer to use passive voice; I believe this practice is not recommended anymore. Excessive use of connecting statements, such as "on the other hand", "conversely", "Therefore", "In conclusion". Please omit the ones that are not absolutely necessary. Title can be a bit punchier.

The language style was not improved too much in this first submission, but we agree to avoid the passive voice and try to reduce the connecting statements. We are going to consider a Proofreading of the work. Title: We know that is a bit general and not specific.

Abstract It is long and it lacks providing a motivation and a take home message. For example, first two sentences of the abstract sounds similar. There are also a few sentences without any information, e.g. "The results obtained from the application of the two methodologies have been discussed.". It also does not involve any clear take home message at the end, as expected.

Abstract REPLY:

Again, we agree with you that is general and should be more focused on the results of the paper.

Introduction The introduction is really hard to follow, it does not develop logically leading to the research questions of - Are these approaches sufficient for a complete description of triggering factors? - Can rainfall analysis be improved considering also other meteorological variables, which could better describe the rainfall events and the linked consequences? Why "other meteorological variables" are not mentioned before? Which approaches could be an alternative? These questions are not returned at the discussion one more time. A topic in a paragraph is returned after a few paragraphs again that confuses the reader. There are a few references, e.g. Guzzetti; Rosi;

Gao, that appear frequently in different paragraphs. Authors should review the global literature carefully and reformulate the introduction leading to the research questions also beyond their study area. Line 47: "However, for shallow landslides: : :" I can't understand this sentence. Line 52: "These thresholds data are calibrated looking at the past events occurred in the area and directly correlated with the nearest rain gauge measures (Rappelli, 2008)." Rainfall intensity might vary considerably during an event, gauge data might miss this variation. We see this especially when rainfall radar estimates are compared with the gauge observations. Authors should also mention this effect in the same paragraph. Line 58: ": : :rainfall thresholds have been widely used in different parts of the world.", but the authors refer usually to the studies from Italy, if I am not wrong. Line 74-82: Similar statements are mentioned in the Abstract as well as early in the Introduction, please refrain using same statements again and again.

Introduction REPLY:

Looking again at our introduction it seems that we are turning around the paper topic, but we do not explain clearly the question we have raised. Here the literature should be extended to formulate the problem we are analyzing in a clear and linear way, posing the questions and then start with the presentation of our own strategy to solve it.

We agree with the LINE comments: LINE 47: Deep seated landslides or big landslide have a complex triggering mechanism where geology and local morphology have an important role that can overcome the meteorological triggering effects. This is not the case for shallow movements that are related to the oversaturation of superficial terrain that is highly dependent on rainfalls triggering. So that Rainfall can be assumed as a predictor for failure. LINE 52: We agree with this suggestion and we will include. LINE 58: It is true because a lot of studies has been carried out in Italy. Also in other part of the world such as JAPAN and California these approach have been studied. We can include these citations. LINE 74-82: we can skip them in order to not refrain the same statement.

Data, Methods and Models Authors should consider using dedicated chapters as "Data" and "Methods". The method part explains how the computation works but lacks information about the meaning of the results for the current study. Line 96: ": : :estimated in 2 billion of euros." reference missing. Line 98: ": : :glacier melting increased by high-altitude summer temperatures." reference missing. Line 143: ": : :extratropical cyclone structures" Different type of rainfalls that effect the regions might be important for the entire study. Authors should consider providing more info about these effect in the introduction. Line 146: ": : :the traditional rainfall approach and the meteorological reanalysis approach." They are not mentioned before. Line 154: ": : :by several authors.' Who are they and how they mention it? Line 187: ": : :extratropical cyclone (EC), as described in Figure 2" I am not sure whether I can see the message in the figure.

Data and Methods REPLY: Data and methods will be presented in two different sections, with a better explanation of the attended results. LINE 96 and 98: we missed them, but we will include. LINE 143: we will include a brief explanation of EC in the Introduction as said before. LINE 146: again, this could be included in the introduction. LINE 154: reference is missing, we should include it. LINE 187: it should be more explained within the Caption of fig. 2 where are evidenced the typical structure of EC cyclones.

Results and Discussion It is hard to follow what is the new result and what is the discussion point. Please consider using a dedicated section for each, "Results", "Discussion". Line 236: ": : :rain rate ÃřĚŹIRĚĞ ij: : :" I guess rain intensity Line 255: ": : :possible indicator of the magnitude of the hydrogeological events..." : : :of: : :of: : : Line 262: ": : :does not permit: : :", do you mean "hinders", I recommend authors to use either negative prefixes and suffixes, or negative verbs directly instead of negative verb conjugation throughout the manuscript. For example, in line 281: ": : :it does not exist a unique method for the magnitude assessment: : :"¡': : :a unique method lacks that assess the magnitude: : :"

[Figure]

Result and Discussion REPLY: We agree that information stored in the result section should be discussed in the dedicated section. In fact, comparisons among the Rainfall Analysis approach and the Meteorological Analysis are crucial to motivate our study. We agree with the LINE comments LINE 236: Yes, Rain Intensity LINE 262-281: We agree with the English style

Conclusion The section is dedicated to summarize the applied analyses, I could not really find a clear take home message that evolved from the results and discussion. Line 384: What are those "hydrogeological issues"? After reading the entire manuscript, I am still not sure about this term? Are the authors refer to hydrogeological evets, such as rainfall induced landslides? Or are there other events?

Conclusion REPLY: We think that with a more structured discussion the conclusion part will be easier to write in a more incisive style. What we want to say is this: Rainfall Analysis can help to identify the triggering events that have caused hydrogeological issues, but we must be aware of the uncertainties around that method. Considering the new possibility coming from meteorological analysis, the definition of the triggering event can be assessed not only considering the rainfall data, but evaluating an index that consider the evolution in space and in time of the entire meteorological event. The latter is not site-specific, and it is physical based but can be applied only for a category of meteorological phenomena (EC). We have tested it and we demonstrated that it can be useful and can be correlated to the magnitude of the hydrogeological issues, giving comparable results with the Rainfall Analysis we carried out. ***LINE 384: for hydrogeological issues we intend all the hydrogeological phenomena that can be triggered mainly by rainfalls, such as shallow landslide, debris flows, flash floods etc.

Tables: Table 1: The caption is repeating the column titles of the table. Figures: There are several bar plots, which does not provide so much information. They may be fit for the appendix, but they are poor for the main body of the manuscript. Authors should consider re-generating figures that provide clear messages. They could consider com-

bining several of the figures in a more creative manner. Figure 1: X and Y labels and ticks are missing. Fonts are larger than in the main text. Figure 2: What is the purpose of the arrows? Figure 4: What is mid-latitude-and highlands climate? Figure 6 and figure 7: I cannot understand the message of these two, especially the figure 7 shows nearly constant Geostrophic velocity (_42+-5 km/h) Figure 8: Location of the Alps are different in each subplot; Lat Lon data is missing the numbers on the isolines are too small to read.

Tables and Figures REPLY: We have tried to be as clear as possible, but we know that synthesis is appreciated in the captions. However, we agree that a more creative representation of the figures should be achieved to facilitate the message we want to show to the reader. TAB 1: we make it shorter FIG 1: we should put a georeferentiation of the map and uniform the text height FIG 2: the arrow indicates the southerly flow that characterize the EC structure. Southerly flow (in Italy called Scirocco) is a moist air flow that it is responsible of the torrential rainfalls that are triggered around the Alps range when it is forced to rise by the mountains. FIG 3: Mid latitude and Highlands represent the classification proposed by Guzzetti for dividing the triggered hydrogeological phenomena respect to the climate location. Mid-Latitude refers to the Temperate climate (Italy is in this) and the Highlands are related to the High Mountains area (where ice and frost are present. In our region, central Alps both environments are present, so we have considered both. FIG 6: represents the two indicators m1 and m2 and their average for the different event type. FIG 7: Geostrophic velocity is a rough indicator of the intensity of the EC structure. Generally, strong ECs show strongest velocities that ranges around 40 km/h. It is a confirmation that all the event we analysed are associated with EC structures. We have presented them because have a key role in determining the value of the SLPT index. FIG 8: The Alps Range was added just as a reference, but we can improve its representation.
* * *

---

## Author Comment (AC4) · 29 Dec 2020

REPLY TO REFEREE 2 We would like to thank the reviewer for the time and effort he/she put into thoroughly reviewing this manuscript. We believe that the comments are constructive and would lead the considerable improvement of our work.
The manuscript deals with an interesting attempt to strengthen the estimation of extreme hydrological conditions, which led to landslide and flash flood phenomena in the Sondrio Province (northern Italy) in the period 1951-2019, by linking them to estimations of magnitude and return period (temporal probability) as well as meteorological conditions occurred at the continental scale. In the opinion of this reviewer, such a methodological effort is certainly to be appreciated because tending to reduce uncertainties of a single approach, such as empirical rainfall thresholds for shallow landslide triggering and hydrological probabilistic models. Specifically, starting from an inventory of principal landslide and flash flood phenomena collected for the study area, authors analyzed triggering hydrological factors by three methods. The first is based on the comparison of intensity/duration of rainfall events, which led to landslide and flash flood phenomena, to empirical rainfall thresholds for shallow landslides known by the literature for the region studied and worldwide. The second is the estimation of return periods for triggering rainfall conditions by a known regional probabilistic model (De Michele et al., 2005). The third is the analysis of meteorological conditions which lead to extreme rainfall events, based on data taken from the National Centres of Environmental Prediction (NCEP) (Kalnay et al., 1996; MeteoCiel, 2020; NOAA, 2020). Notwithstanding the challenging and innovative premises, the manuscript presents several conceptual points of weakness which are described below in form of both general and specific comments.

GENERAL COMMENTS: 1) Rainfall thresholds considered, known by the literature, are related to shallow landslides, instead Authors compare to them also deep-seated phenomena (1987 deep-seated Val Pola landslide) and flash floods. Therefore, the comparison appears too heterogeneous and would need a motivation.

R:Probably is not sufficiently specified but the intent is to try to focus mainly on shallow landslides that are generally more influenced by meteorological condition. We consider your advice and provide a better explanation for this topic.

2) The procedure used for assessing the magnitude ranking (Fig. 6), by taking into account of both return period and areal extent of meteorological phenomena, appears

questionable for being based on the mean of normalized value. By a conceptual point of view, considering the mean value is allowed for the same variable, not for different variables. Very likely, the normalized product of return period and areal extent of meteorological phenomena would incorporate more consistently the magnitude at local scale (return period) and the areal extension.

R:We have worked out about that topic and we found the same inconsistency. In fact, affected area and rainfall return period contain 2 important information that are complementary to explain an intensity of a rainfall triggered event. Therefore, the product of the 2 normalized index seems to be more appropriate. We will ri-elaborate it.

3) Parts of the paragraph 3.2.1 regards general methodological aspects and preceding knowledge, therefore they appear more suitable for the methods section rather than the results one.

R:We will consider to move it in the Method section.

4) Nothing is given about the regional probabilistic model (De Michele et al., 2005), which has been considered to the estimation of return periods. In the opinion of this reviewer, it's an important point to explain aspects related to the regional probabilistic model adopted.

R:We will add further details about this part because is necessary also in the description of Return Period evaluation.

5) Authors should motivate with a greater emphasis the possible applications of their findings in the field of landslide and flood hazard assessment as well as early warning systems.

R:This part is rather fundamental. Our intent is to try to built a bridge among two different discipline that sometimes find the same difficulties around the interpretation hydrogeological event intensities. We will built a paragraph in order to better motivate our study.

6) English language should be revised. The adjective "hydrogeological", used extensively throughout the text, appears not suitable to indicate landslide and flood phenomena. It is recommended to substitute this term. R:We will provide an extensive English editing after during the article revision.

Block 20. The sentence is not clear and it should be rewritten. R:OK Block 30. The use of the term "back analysis" is questionable because it is commonly used in the geotechnics field for inverting a slope stability analysis and estimating shear strength of geological materials involved in landsliding. In this case the analyses carried out are just re-examinations of past landslide and flash flood events. R:OK Block 40. Substitute "deep landslide" with "deep-seated landslide". R:That is more correct. Block 80. Substitute "will be" with "is". R:OK Block 90. "Old debris" is not a geological term. Maybe, just debris could be better, otherwise the age should be indicated more clearly (e.g. Pleistocene). R:That is more precise. Table 1: A column indicating the mean intensity should be considered. The definition of Extremely Localized (EXTL) and Diffuse (DIF) could be substituted with Localized (L) and Areal (A). R:That is more synthetic. Blocks 250-255 and Fig. 4. To consider the vertical distance between the curve and the critical event point appears conceptually incorrect due the possibility that the curve itself (I/D rainfall thresholds) indicates points with different return period. Authors should verify and discuss this point. R:We will consider all these useful advices during the revision of the paper.
* * *

---

## Author Response (AR1)

REPLY TO REFEREE 1

Title: Meteorology triggering factors analysis for rainfall induced hydrogeological events
in alpine region

We are kindly grateful for your accurate revision of our work. We have really appreciated your hints and suggestions and in this brief reply we are going to discuss them.

Overview and general comments:
*Authors studies the past meteorological (mainly rainfall) conditions that lead to certain hydrogeological events. They applied a systematic analysis, and fairly self-criticize their results. The study is relevant for particularly landslide researchers that use rainfall I-D curves. Although the technical framework of the manuscript seems sound and robust, the presentation of the work is rather poor in the current state. Overall, I got the impression that this manuscript is not properly edited before submission. There are several tiny mistakes in the text, which makes it hard to follow. With proper editing the manuscript will get considerably shorter and understandable. This manuscript deserves publication, after considerable changes in its current presentation.*

We are aware that the work is not properly ready for a direct submission because several topics we have analysed should be more integrated. We have improved the presentation of our work in this sense.

MAJOR COMMENTS

*Authors evaluate/discuss the rainfall I-D curves only based on "false negatives", but ignore "false positives".*

We have discussed this point in the text and we have extended the discussion about the issues about the use of I-D thresholds. We have stated that not only I-D parameters are sufficient for correcting spot the events but also other "background elements" such as soil moisture and antecedent rainfall may perturb the detecting performance of these curves. False positives may happen but, in our case,, we have analysed "true positive" events and we have demonstrated that, depending on selected threshold, the critical events detection may sensibly change.

*A certain rainfall might remain above the I-D curve of Guzzetti et al. (2007) or Mid-latitude Climate without triggering any landslide as well. The authors avoid discussing their results on a broader scale beyond their study area.*

We have concentrated on a specific mountain area, but we have in mind that I-D thresholds issues may be founded also in other similar areas.

*They rather give an event summary in the analyses. They should extend their discussion and explain what do we learn from this study that is valid also for other high relief areas?*

We have commented extensively this part in the Results, Discussion and in Conclusion sections.

MINOR COMMENTS

*There are wrong use of words and tenses throughout the manuscript. It should be proof read. Authors prefer to use passive voice; I believe this practice is not recommended anymore. Excessive use of connecting statements, such as "on the other hand", "conversely",*

*"Therefore", "In conclusion". Please omit the ones that are not absolutely necessary.*
*Title can be a bit punchier.*

The language style was not improved too much in this first submission, but we agree to avoid the passive voice and try to reduce the connecting statements. We have considered a First Proofreading of the work. We have improved the language style and made the style more fluent.

Title: We know that is a bit general and not specific. We have modified in order to clarify what we have done in our work.

Abstract
*It is long and it lacks providing a motivation and a take home message. For example,*
*first two sentences of the abstract sounds similar. There are also a few sentences*
*without any information, e.g. "The results obtained from the application of the two*
*methodologies have been discussed.". It also does not involve any clear take home*
*message at the end, as expected.*

Abstract: Again, we agree with you that is general and should be more focused on the results of the paper. We have improved the take home message and we have shortened.

Introduction
*The introduction is really hard to follow, it does not develop logically leading*
*to the research questions of - Are these approaches sufficient for a complete description*
*of triggering factors? - Can rainfall analysis be improved considering also*
*other meteorological variables, which could better describe the rainfall events and the*
*linked consequences? Why "other meteorological variables" are not mentioned before?*
*Which approaches could be an alternative? These questions are not returned at*
*the discussion one more time.*
*A topic in a paragraph is returned after a few paragraphs again that confuses the*
*reader. There are a few references, e.g. Guzzetti; Rosi; Gao, that appear frequently*
*in different paragraphs. Authors should review the global literature carefully and reformulate*
*the introduction leading to the research questions also beyond their study*
*area.*
*Line 47: "However, for shallow landslides: : :" I can't understand this sentence.*
*Line 52: "These thresholds data are calibrated looking at the past events occurred in*
*the area and directly correlated with the nearest rain gauge measures (Rappelli, 2008)."*
*Rainfall intensity might vary considerably during an event, gauge data might miss this*
*variation. We see this especially when rainfall radar estimates are compared with the*
*gauge observations. Authors should also mention this effect in the same paragraph.*
*Line 58: ": : :rainfall thresholds have been widely used in different parts of the world.",*
*but the authors refer usually to the studies from Italy, if I am not wrong.*
*Line 74-82: Similar statements are mentioned in the Abstract as well as early in the*
*Introduction, please refrain using same statements again and again.*

Introduction:

Looking again at our introduction it seems that we are turning around the paper topic, but we do not explain it clearly the question we have raised. Here the literature should be extended to formulate the problem we are analysing in a clear and linear way, posing the questions and then start with the

presentation of our own strategy to solve it. In the new version we have improved this part digging in detail the current methodologies considered for rainfall triggering interpretation.

We agree with the LINE comments:

LINE 47: Deep seated landslides or big landslide have a complex triggering mechanism where geology and local morphology have an important role that can overcome the meteorological triggering effects. This is not the case for shallow movements that are related to the oversaturation of superficial terrain that is highly dependent on rainfalls triggering. So that Rainfall can be assumed as a predictor for failure.

LINE 52: We agree with this suggestion and we will include.

LINE 58: It is true because a lot of studies has been carried out in Italy. Also in other part of the world such as JAPAN and California these approach have been studied. We can include these citations.

LINE 74-82: we can skip them in order to not refrain the same statement.

Data, Methods and Models
*Authors should consider using dedicated chapters as "Data" and "Methods". The method part explains how the computation works but lacks information about the meaning of the results for the current study.*
*Line 96: ": : :estimated in 2 billion of euros." reference missing.*
*Line 98: ": : :glacier melting increased by high-altitude summer temperatures." reference missing.*
*Line 143: ": : :extratropical cyclone structures" Different type of rainfalls that effect the regions might be important for the entire study. Authors should consider providing more info about these effect in the introduction.*
*Line 146: ": : :the traditional rainfall approach and the meteorological reanalysis approach." They are not mentioned before.*
*Line 154: ": : :by several authors.' Who are they and how they mention it?*
*Line 187: ": : :extratropical cyclone (EC), as described in Figure 2" I am not sure whether I can see the message in the figure.*

Data and Methods:

Data and methods will be presented in two different sections, with a better explanation of the attended results. We have included Data and Materials section and the Model and Methods section in order to better distinguishing the database investigated form the methodology applied.

LINE 96 and 98: we missed them, but we will include.

LINE 143: we will include a brief explanation of EC in the Introduction as said before.

LINE 146: again, this could be included in the introduction.

LINE 154: reference is missing, we should include it.

LINE 187: it should be more explained within the Caption of fig. 2 where are evidenced the typical structure of EC cyclones.

Results and Discussion
*It is hard to follow what is the new result and what is the discussion point. Please*

*Line 236: ": : :rain rate ðˈIRˇij: : :" I guess rain intensity Line 255: ": : :possible indicator of the magnitude of the hydrogeological events..." : : :of: : :of: : : Line 262: ": : :does not permit: : :", do you mean "hinders", I recommend authors to use either negative prefixes and suffixes, or negative verbs directly instead of negative verb conjugation throughout the manuscript. For example, in line 281: ": : :it does not exist a unique method for the magnitude assessment: : :"!": : :a unique method lacks that assess the magnitude: : :"*

Result and Discussion:

We agree that information stored in the result section should be discussed in the dedicated section. In fact, comparisons among the Rainfall Analysis approach and the Meteorological Analysis are crucial to motivate our study. We have split Results from Discussion as well.

We agree with the LINE comments

LINE 236: Yes, Rain Intensity

LINE 262-281: We agree with the English style

Conclusion
*The section is dedicated to summarize the applied analyses, I could not really find a clear take home message that evolved from the results and discussion. Line 384: What are those "hydrogeological issues"? After reading the entire manuscript, I am still not sure about this term? Are the authors refer to hydrogeological evets, such as rainfall induced landslides? Or are there other events?*

Conclusion:
We think that with a more structured discussion the conclusion part will be easier to write in a more incisive style. What we want to say is this:

Rainfall Analysis can help to identify the triggering events that have caused hydrogeological issues, but we must be aware of the uncertainties around that method. Considering the new possibility coming from meteorological analysis, the definition of the triggering event can be assessed not only considering the rainfall data, but evaluating indexes that consider the evolution in space and in time of the entire meteorological event. The latter is not site-specific, and it is physical based but can be applied only for a category of meteorological phenomena (EC). We have tested it and we demonstrated that it can be useful and can be correlated to the magnitude of the hydrogeological issues, giving comparable results with the Rainfall Analysis we carried out.

In the conclusion we have remarked this fact we highlighted more the positive outcomes of our study.

***LINE 384: for hydrogeological issues we intend all the hydrogeological phenomena that can be triggered mainly by rainfalls, such as shallow landslide, debris flows, flash floods etc.

Tables:
*Table 1: The caption is repeating the column titles of the table.*
*Figures: There are several bar plots, which does not provide so much information.*
*They may be fit for the appendix, but they are poor for the main body of the manuscript.*
*Authors should consider re-generating figures that provide clear messages. They could consider combining several of the figures in a more creative manner.*

*Figure 1: X and Y labels and ticks are missing. Fonts are larger than in the main text.*
*Figure 2: What is the purpose of the arrows?*
*Figure 4: What is mid-latitude-and highlands climate?*
*Figure 6 and figure 7: I cannot understand the message of these two, especially the*
*figure 7 shows nearly constant Geostrophic velocity (_42+-5 km/h)*
*Figure 8: Location of the Alps are different in each subplot; Lat Lon data is missing the*
*numbers on the isolines are too small to read.*

Tables and Figures:

We have tried to be as clear as possible, but we know that synthesis is appreciated in the captions. However, we agree that a more creative representation of the figures should be achieved to facilitate the message we want to show to the reader.

TAB 1: we make it shorter

FIG 1: we should put a georeferentiation of the map and uniform the text height

FIG 2: the arrow indicates the southerly flow that characterise the EC structure. Southerly flow (in Italy called Scirocco) is a moist air flow that it is responsible of the torrential rainfalls that are triggered around the Alps range when it is forced to rise by the mountains.

FIG 3: Mid latitude and Highlands represent the classification proposed by Guzzetti for dividing the triggered hydrogeological phenomena respect to the climate location. Mid-Latitude refers to the Temperate climate (Italy is in this) and the Highlands are related to the High Mountains area (where ice and frost are present. In our region, central Alps both environments are present, so we have considered both.

FIG 6: represents the two indicators m1 and m2 and their average for the different event type. (erased)

FIG 7: Geostrophic velocity is a rough indicator of the intensity of the EC structure. Generally, strong ECs show strongest velocities that ranges around 40 km/h. It is a confirmation that all the event we analysed are associated with EC structures.  We have presented them because have a key role in determining the value of the SLPT index. (erased partially)

FIG 8: The Alps Range was added just as a reference, but we can improve its representation. (erased)

Form the previous version we have erased some usefulness figures and tables in order to be more concise in results explanation.

RELEVANT CHANGES

Improved the style and the organization of the paper;

Revision of the I-D methodology and discussed around its possible application in the case study presented;

Revision the methodology of magnitude indexes estimation: definition if the new magnitude index MI based on database data, frequency-magnitude theory and statistical interpretation;

Improved the result and discussion sections.

REPLY TO REFEREE 2

*We would like to thank the reviewer for the time and effort he/she put into thoroughly reviewing this manuscript. We believe that the comments are constructive and would lead the considerable improvement of our work.*
The manuscript deals with an interesting attempt to strengthen the estimation of extreme hydrological conditions, which led to landslide and flash flood phenomena in the Sondrio Province (northern Italy) in the period 1951-2019, by linking them to estimations of magnitude and return period (temporal probability) as well as meteorological conditions occurred at the continental scale. In the opinion of this reviewer, such a methodological effort is certainly to be appreciated because tending to reduce uncertainties of a single approach, such as empirical rainfall thresholds for shallow landslide triggering and hydrological probabilistic models. Specifically, starting from an inventory of principal landslide and flash flood phenomena collected for the study area, authors analyzed triggering hydrological factors by three methods. The first is based on the comparison of intensity/duration of rainfall events, which led to landslide and flash flood phenomena, to empirical rainfall thresholds for shallow landslides known by the literature for the region studied and worldwide. The second is the estimation of return periods for triggering rainfall conditions by a known regional probabilistic model (De Michele et al., 2005). The third is the analysis of meteorological conditions which lead to extreme rainfall events, based on data taken from the National Centres of Environmental Prediction (NCEP) (Kalnay et al., 1996; MeteoCiel, 2020; NOAA, 2020).

Notwithstanding the challenging and innovative premises, the manuscript presents several conceptual points of weakness which are described below in form of both general and specific comments.

GENERAL COMMENTS:

1) Rainfall thresholds considered, known by the literature, are related to shallow landslides, instead Authors compare to them also deep-seated phenomena (1987 deep-seated Val Pola landslide) and flash floods. Therefore, the comparison appears too heterogeneous and would need a motivation.

*Probably is not sufficiently specified but the intent is to try to focus mainly on shallow landslides that are generally more influenced by meteorological condition. We consider your advice and provide a better explanation for this topic.*

2) The procedure used for assessing the magnitude ranking (Fig. 6), by taking into account of both return period and areal extent of meteorological phenomena, appears questionable for being based on the mean of normalized value. By a conceptual point of view, considering the mean value is allowed for the same variable, not for different variables. Very likely, the normalized product of return period and areal extent of meteorological phenomena would incorporate more consistently the magnitude at local scale (return period) and the areal extension.

*We have worked out about that topic and we found the same inconsistency. In fact, affected area and rainfall return period contain 2 important information that are complementary to explain an intensity of a rainfall triggered event. Therefore, the product of the 2 normalized index seems to be more appropriate. We have ri-elaborate it in a stronger way considering the frequency-magnitude theory and some statistical assumptions.*

3) Parts of the paragraph 3.2.1 regards general methodological aspects and preceding knowledge, therefore they appear more suitable for the methods section rather than the results one.

*We have rewritten this part.*

4) Nothing is given about the regional probabilistic model (De Michele et al., 2005), which has been considered to the estimation of return periods. In the opinion of this reviewer, it's an important point to explain aspects related to the regional probabilistic model adopted.

*We added further details about this part because is necessary also in the description of Return Period evaluation and extended the presentation both in the Methods and in Results.*

5) Authors should motivate with a greater emphasis the possible applications of their findings in the field of landslide and flood hazard assessment as well as early warning systems.

*This part is rather fundamental. Our intent is to try to built a bridge among two different discipline that sometimes find the same difficulties around the interpretation hydrogeological event intensities. We built a paragraph in order to better motivate our study in the discussion section.*

6) English language should be revised. The adjective "hydrogeological", used extensively throughout the text, appears not suitable to indicate landslide and flood phenomena. It is recommended to substitute this term.

*We have provided an first English editing during the article revision. The Final English editing will be carried out after the acceptance of the paper.*

Block 20. The sentence is not clear and it should be rewritten.

Block 30. The use of the term "back analysis" is questionable because it is commonly used in the geotechnics field for inverting a slope stability analysis and estimating shear strength of geological materials involved in landsliding. In this case the analyses carried out are just re-examinations of past landslide and flash flood events.

*We have changed this into "reanalysis"*

Block 40. Substitute "deep landslide" with "deep-seated landslide". *That is more correct.*

Block 80. Substitute "will be" with "is".

Block 90. "Old debris" is not a geological term. Maybe, just debris could be better, otherwise the age should be indicated more clearly (e.g. Pleistocene).

*That is more precise and we have erased this therm.*

Table 1: A column indicating the mean intensity should be considered. The definition of Extremely Localized (EXTL) and Diffuse (DIF) could be substituted with Localized (L) and Areal (A).

*That is more synthetic but we preferred to keep that notation in order to avoid mismatches with other therms used in the texts: L = low pressure and A is similar to AA (Affected Area value)*

Blocks 250-255 and Fig. 4. To consider the vertical distance between the curve and the critical event point appears conceptually incorrect due the possibility that the curve itself (I/D rainfall thresholds) indicates points with different return period. Authors should verify and discuss this point.

*We have consider all these useful advices during the revision of the paper and we rewritten this part in result and discussion section.*

RELEVANT CHANGES

Improved the style and the organization of the paper;

Revision of the I-D methodology and discussed around its possible application in the case study presented;

Revision the methodology of magnitude indexes estimation: definition if the new magnitude index MI based on database data, frequency-magnitude theory and statistical interpretation;

Improved the result and discussion sections.

---

## Editor Decision (ED1)

[revised manuscript text omitted]

---

## Author Response (AR2)

Referee n°1

*Dear Referee n°1*

*We are kindly grateful for your accurate revision of our work. We have appreciated your hints and suggestions and we thank you for your contribution to the improvement of our work.*

*Best regards*

*The Authors*

Referee n°2

*Dear Referee n°2*

*We are kindly grateful for your accurate revision of our work. We have appreciated your hints and suggestions and we thank you for your contribution to the improvement of our work.*

*Best regards*

*The Authors*

The manuscript has been reworked and improved in several parts. This reviewer recognizes that authors have accomplished the most important observations, such as the reformulation of the MI Index and to not consider, in the I/D graph, the distance between points and the reference curve as a direct indicator of the return period.

Notwithstanding these advances, this reviewer indicates some remarks whose accomplishment is needed for definitive improvements of the manuscript.

The first one is terminological. Authors still use the adjective "hydrogeological" to indicate indistinctly flood and rainfall-induced landslide phenomena, by simply translating the equivalent term from the Italian, which is itself improper and just popular. As it is well known, "hydrogeological" is used in the scientific literature to indicate groundwater flow phenomena and their relationships with hosting rocks and soils. The unifying term (adjective) for indicating floods and rainfall-induced landslides could be "hydrogeomorphological".

*We have substituted the word "hydrogeological" in "geo-hydrological" as suggested also by Editor.*

The second regards the Fig. 10b in which the inversion of the dependent variable (SLPT Index) with the independent one (MI Index) is suggested to make the inherent empirical relationship usable for predicting MI Index by the SLPT Index.

*We have changed the figure according to this indication, showing the estimation of the MI in function of the SLPT.*

The third is an accurate revision of the English language which in several parts of the manuscript still presents grammar errors and construction of sentences too similar to the Italian language.

*We have corrected some grammar errors, typos and improved the readability of overall the paper*